# Hierarchy-Agnostic Unsupervised Segmentation: Parsing Semantic Image Structure

**Simone Rossetti**[1,2]     **Fiora Pirri**[1,2]
[1]DIAG, Sapienza University of Rome     [2]DeepPlants
{rossetti,pirri}@diag.uniroma1.it
{simone,fiora}@deepplants.com

## Abstract

Unsupervised semantic segmentation aims to discover groupings within images, capturing objects' view-invariance without external supervision. Moreover, this task is inherently ambiguous due to the varying levels of semantic granularity. Existing methods often bypass this ambiguity using dataset-specific priors. In our research, we address this ambiguity head-on and provide a universal tool for pixel-level semantic parsing of images guided by the latent representations encoded in self-supervised models. We introduce a novel algebraic approach that recursively decomposes an image into nested subgraphs, dynamically estimating their count and ensuring clear separation. The innovative approach identifies scene-specific primitives and constructs a hierarchy-agnostic tree of semantic regions from the image pixels. The model captures fine and coarse semantic details, producing a nuanced and unbiased segmentation. We present a new metric for estimating the quality of the semantic segmentation of discovered elements on different levels of the hierarchy. The metric validates the intrinsic nature of the compositional relations among parts, objects, and scenes in a hierarchy-agnostic domain. Our results prove the power of this methodology, uncovering semantic regions without prior definitions and scaling effectively across various datasets. This robust framework for unsupervised image segmentation proves more accurate semantic hierarchical relationships between scene elements than traditional algorithms. The experiments underscore its potential for broad applicability in image analysis tasks, showcasing its ability to deliver a detailed and unbiased segmentation that surpasses existing unsupervised methods.

## 1 Introduction

The advancement of image segmentation has recently taken significant steps forward. On the one hand, the foundation models are trained on increasingly large datasets, such as CLIPseg [57] (CLIP [67]), SAM [48], and SEEM [97], supervised by text and human prompts [92]. On the other hand, there is a rising growth of unsupervised segmentation models. Unsupervised segmentation explores the feature hierarchy by leveraging self-supervised contrastive learning, as in SmooSeg [51], U2Seg [62], CUTLer [87], CuVLER [5], STEGO [32], ACSeg[52], FreeSolo [85], HSG [44], Trans-Fgu [90], DeepCut [3], and others [77, 58, 96, 86, 33, 17]. Unsupervised models discover and localize image categories with no annotation aid and evaluate the quality of the pseudo-masks they predict on the datasets corpora used for testing, such as COCO-Stuff [10] and Cityscapes [21]. Despite exploring human perception more closely than the foundation models, they still rely on the linguistic and conceptual relations between the images and their annotations.

Curated datasets, such as ImageNet [50], PascalVOC [27], or MSCOCO [53], show an extraordinary number of objects with all their components and particulars not annotated either at the image level

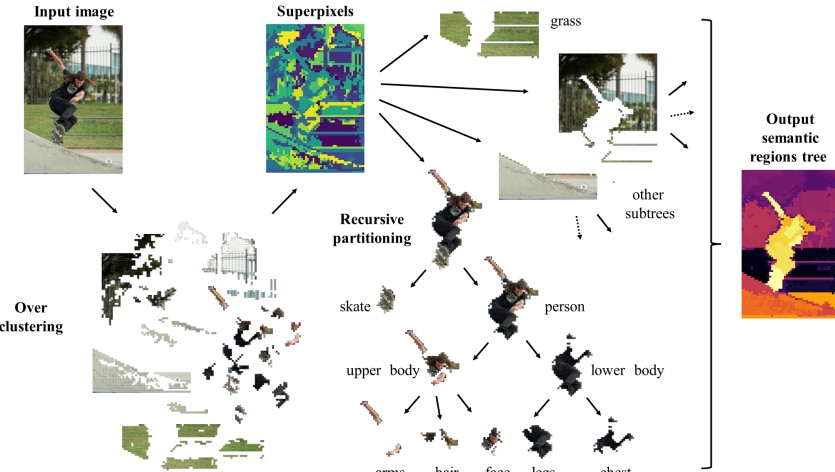

Figure 1: **Hierarchy-agnostic unsupervised segmentation.** Finer image parts are generated via over-clustering, each region colour-coded randomly. Our algorithm recursively partitions these parts, grouping them into coarser regions across multiple levels of granularity. The resulting tree represents an unsupervised hierarchical semantic segmentation. The arrangement of regions in the tree reflects their semantic distance, which is colour-coded in the heat map shown on the right.

or densely. Why annotate this and not that? Annotation prejudice creates a bias towards a subset of the scene. Unsupervised learning, in contrast, has the potential to generate richer representations that are not restrained by annotation decisions. Without juggling annotations, unsupervised features (e.g. [64]) nearly mirror visual perception discovering scene parts and details; indeed, feature cues live in nested context levels and are not necessarily verbalisable [76, 66], as opposed to the Gestalt holistic view [83].

Following previous research [44, 19, 3, 94], our key idea is that the natural hierarchical structure of visual scenes is an essential attribute that we can actively pursue in unsupervised segmentation. We approach unsupervised semantic segmentation as an unsupervised pixel-wise feature learning problem, discretising the hierarchical semantic knowledge yielded by self-supervised learning. Our approach makes no assumption about the number of semantic granularity levels and the number of partitions in each level as in [44]. We generate robust hierarchical segmentation for every image, solely relying on hierarchical clustering in feature space, see Figure 1. This new method achieves unsupervised segmentation by leveraging relationships between concepts hidden in the latent space of self-supervised models, across multiple levels of semantic granularity.

We propose a simple algebraic methodology based on a vast literature [20, 6, 61, 81, 75, 91], that unsupervisedly segments the scene parts. The method guarantees the construction of natural, scene-dependent classes of primitives [55], which can be easily used in unsupervised segmentation without surrendering to their a priori definitions. Our contributions are:

1. We introduce a deep recursive spectral clustering that maximises an unbiased semantic similarity measure at multiple granularity levels. Exposing our method to any generic group of images, we show that it results in hierarchical unsupervised semantic segmentation.

2. We introduce new metrics for estimating the quality of the semantic segmentation of the elements discovered on the different levels of the hierarchy, called *Normalised Multigranular Covering* (NMCovering) and *Normalised Hierarchical Covering* (NHCovering).

3. We integrate the method into various self-supervised learning models to enhance its flexibility for benchmarking purposes, making it suitable as a downstream task through unsupervised segmentation by inferring all scene parts in the images of any given dataset.

## 2   Related works

**Self-supervised representation learning for unsupervised segmentation.** Self-supervised learning (SSL) is about learning representations of real-world data without human supervision [14, 36, 11,

64]. Contrastive learning (CL) [18, 63] is the most prominent method in SSL, maximising feature similarities between an image and its affine transformations while minimising similarity between randomly sampled images. Most unsupervised segmentation methods use learning representations to feed self-supervised features to graphs for clustering [58, 88, 87, 3]. Alternatively, features are used for building distillation strategies [32, 96, 51], or for designing inductive priors forcing some consistency property [51, 17, 73]. SSL representations implicitly define a distance metric in the latent space, though learning a metric space is not their primary objective.

**Unsupervised semantic segmentation.** Unsupervised semantic segmentation labels pixels that belong to specific elements in the scene, grouping all objects of the same semantic type under a single label, without supervision.

The earlier approaches, such as [41, 47, 17, 86, 77, 33] had not yet available powerful SSL features like [11], though used CL principles. PiCie [17] used equivariance to transformations; IIC [41] resorted to invariant information clustering, maximising mutual information, to find commonality in objects, and analogously did InfoSeg [33]. CLD [86] integrated local clustering into contrastive learning; [77] used saliency to find the image foreground and guide CL of pixel embedding.

The availability of high-level unsupervised features triggered new strategies. DINO self distillation [11] inspired both STEGO [32] and SmooSeg [51]. STEGO trains a segmentation head by distilling the feature correspondences to form compact clusters. SmooSeg uses the smooth prior over semantically coherent regions as a supervision cue to generate semantic maps. However, many methods suffer from the background problem and elaborate on DINO's attention to obtaining foreground objects, such as FreeSolo [85] and [93]. Also, TransFGU [90] focuses on a top-down object-centric approach to generate pixel pseudo labels according to GradCAM [72]. Spectral clustering, as introduced to machine learning by [74], is considered in [88, 58, 87, 52, 68]. In particular, CutLER [87] applies NCut [74] iteratively to a masked similarity matrix to discover foreground elements of the scene. ACSeg [52] uses the affinity matrix to discover concept similarities. SemPart [68] considers the foreground a single object saliency mask and applies graph regularisation. In [73], patch-level contrasting learning leverages global hidden positives to learn semantic consistency.

As grouping is the common denominator, all the mentioned methods suffer from deciding the correct level of granularity. Some methods resort to an *object-centric* bias, such as [77, 93], CAMs [90], hinting self-attention [96, 79], fixed-size flat image partitioning [58]. Others, such as [96], and [46, 73], resort to a *scene-centric* prior assumption.

**Unsupervised parts discovering.** Despite hierarchically discovering parts has a long history in computer vision, it has only recently recovered and connected to unsupervised part segmentation. The first input came from [31], analysing the hierarchical nature of deep learning features. One of the first approaches was SCOPS [39] using dense self-supervised contrastive loss to discover foreground parts of single objects. In [94], the authors introduce self-supervised primitive hierarchical grouping. They leverage a boundary strength map (OWT-UCM [4]) on relatively few images to learn from a large data set. The approach formulates an ultrametric map defining a region hierarchy. Similarly, HSG [44] leverages region boundaries to obtain multi-level segmentation. HSG is unsupervisedly trained from scratch, performing pixel grouping with dense contrastive learning across different granularity levels. In [19], $K$ fixed parts are discovered via an average part descriptor and by forcing consistency using the equivariance of affine and photometric transformations. Leopart [96] follows DINO self-distillation to classify pixels, obtaining detailed scene parts, further clustered via community detection. Finally, DeepCut [3] approaches unsupervised semantic part segmentation using both spectral clustering with NCut and GNN convolution, constructing a patch-wise correlation [7] matrix from DINO features.

## 3 Method

We present a flexible, unsupervised method for segmenting natural images, automatically creating data structures that organize pixels by their semantic coherence across multiple levels of detail without relying on predefined hierarchies or labels. This method is designed to segment images from coarse regions to finer parts, providing an intuitive representation of visual content; see Figure 1.

For instance, consider an urban street view. At a high level, it consists of elements such as the *sky*, a *road*, *buildings*, and *vehicles*. Among the vehicles, there might be a *bus* or a *car*, which can be further decomposed into parts like the *body shell*, *wheels*, and other visible components.

Our approach segments an image $I \in \mathbb{R}^{3 \times M \times N}$ into a hierarchy-agnostic tree $T$ of semantic regions, with each pixel in the image assigned to a leaf node. Our model is a function $f : I \to T$, represented by a deep neural network, which maps an image to its semantic regions. Notably, this is achieved in a fully unsupervised manner; we incorporate mechanisms that guide $f$ to produce a meaningful decomposition of the image, even without labelled examples.

## 3.1 Overview of the Approach

Our method discovers similar parts in an image by recursively partitioning a graph constructed from deep feature representations of the image. The key idea is to treat self-supervised models for image processing as *codebooks* of a lower-dimensional space, with the extracted feature vectors acting as *codes* that embed semantics of visual concepts.

The primary cue for discovering parts is a deep feature extractor $\phi$, a neural network pre-trained without supervision on a standard benchmark such as ImageNet. We observe that the alignment of codes leads to semantic similarity across multiple levels of granularity. A higher degree of alignment indicates semantic similarity at finer granularity levels, corresponding to indivisible object parts or *primitives*. Conversely, a lower degree of alignment reflects semantic similarity at coarser granularity levels. By discretizing the density of these alignments, we can discover scene and object parts at various levels of detail.

Let $v_i = [\phi(I)]_i \in \mathbb{R}^d$ be the code associated with pixel location $i$ in the image. If pixel $j$ belongs to the same finer part as $i$, their codes should be similar. Conversely, if they belong to different parts, their codes will diverge. We expect this property to be consistent in each image $I$ and should not be affected by the particular object instances.

A straightforward approach might be to cluster these codes using algorithms that minimize a distortion metric in the latent space to identify regions of high feature concentration. However, a critical flaw that can arise when using these methods in high-dimensional space is the presence of many local minima in the cost function. This would require multiple restarts of the iterative algorithms to find a suitable solution, which is impractical due to high complexity. To overcome these limitations, we adopt an efficient method to partition the image into similar regions, avoiding the pitfalls of multiple local minima and the need for iterative restarts in high-dimensional space.

**Graph construction.** We represent the image $I$ as a weighted undirected graph $G = (V, E, w)$, where $V = \{v_i\}_{i=1}^n$ and $n = M \cdot N$. The weight assigned to each edge $(i, j) \in E$ is defined by a scaled and shifted cosine similarity between feature vectors $w_{ij} = w(v_i, v_j) \in [0, 1]$. These weights form the adjacency matrix $W = [w_{ij}] \in [0, 1]^{n \times n}$, the degree matrix $D = \text{diag}[d_i] \in \mathbb{R}^{n \times n}$, where $d_i = \sum_j w_{ij}$ and the normalized graph Laplacian $L = D^{-1/2}(D - W)D^{-1/2}$ [59].

We interpret the edge weights as indicators of the semantic granularity between nodes. Specifically, if $w_{ij} \to 1$, pixels $i$ and $j$ likely belong to the same fine-grained part (primitive). Conversely, if $w_{ij} \to 0$, these pixels are likely to belong to entirely different parts, indicating minimal semantic similarity even at the coarsest level of granularity.

**Similarity perturbation.** In an ideal scenario, at a specific granularity level, $k'$ distinct connected components emerge in $G$, resulting in a binary adjacency matrix $W' \in \{0, 1\}^{n \times n}$ with $k'$ non-zero diagonal blocks — indicating strong intra-component connectivity and no inter-component connections — and normalized Laplacian $L'$. However, in practice, the observed adjacency matrix is not discrete. Instead, $W$ exhibits tightly connected components alongside others with lower connectivity, resulting in a perturbed normalized graph Laplacian $L$. We regard the primitives of the model as affected by a *small* symmetric perturbation $H \in \mathbb{R}^{n \times n}$ incorporating contextual information from coarser levels of semantic granularity, i.e. $L = L' + H$, making the observed adjacency real-valued.

Fortunately, the Davis-Kahan symmetric $\sin \theta$ theorem [23, 91] helps us manage perturbations; see also Appendix A. If the eigenvalues of $L$ exhibit a spectral gap $\delta$, the corresponding eigenspaces of $L$ and $L'$ remain close despite the perturbation $H$. The theorem quantifies this proximity by relating the angle $\theta$ between the eigenspaces of $L$ and $L'$, where $\sin \theta$ is proportional to the norm of $H$ and inversely proportional to $\delta$. By selecting the largest gap, we isolate the part of the spectrum closest to the original graph, guessing the true semantic structure at the specific granularity level.

**Normalized smoothness measure.** We consider a function $g : V \to \mathbb{R}$ that assigns a value $g(v_i)$ to each node $v_i \in V$. Since $|V| = n$, we identify the function $g$ with a vector in $\mathbb{R}^n$. Based on the considerations from [74, 8] (see also Appendix C), we define the normalized smoothness measure of $g$ on the graph $G$ using the functional $S_G : \mathbb{R}^n \to \mathbb{R}^+$ induced by $L$ through the form:

$$S_G(g) = \frac{g^\top (D - W) g}{g^\top D g} = \frac{\sum_{ij} (g(v_i) - g(v_j))^2 w_{ij}}{\sum_i g(v_i)^2 d_i}. \tag{1}$$

We observe that minimizing the functional yields normalized smoothest functions that assign similar values to tightly connected nodes and different values to weakly connected ones while accounting for their importance in the graph — avoiding trivial solutions for low connectivity nodes. Therefore, if $g$ is both normalized and smooth with respect to $G$, then $g(v_i)$ is similar to $g(v_j)$ whenever $v_i$ is similar to $v_j$, where similarity is quantified by the weight $w_{ij}$ and adjusted by the node degree $d_i$.

Given these properties, we treat any function $g$ as a *continuous partition* function on the graph $G$ and evaluate its correctness by a *feature density change* criterion on the data partition, which measures variations in similarity across different regions of the graph.

**Recursive partitioning with perturbation stability.** We propose a recursive partitioning strategy for discovering semantic parts by progressively dividing the graph, starting from the whole and refining it into tightly connected subgraphs. At each recursion level, we examine the subgraph's granularity, identify perturbations — contextual variations affecting node connections — and derive the unperturbed adjacency matrix to reveal finer semantic components. By capturing relationships at multiple levels of detail, this approach yields more nuanced segmentation than methods that partition the entire graph's nodes into a fixed number $k$ of sets.

We aim to find the smoothest normalized functions that best describe the semantic structure of a subgraph $G$ at a specific granularity.[1] We tackle the minimization of Equation (1) as a standard eigenvalue problem [74] using the Rayleigh-Ritz quotient form. This yields the orthonormal eigenvectors $y_i$ corresponding to the smallest eigenvalues $\lambda_i$ of $L$, as guaranteed by the Courant-Fischer theorem:

$$y_i = \text{argmin}_{\|y\|=1, y \perp y_{<i}} y^\top L y, \text{ with } y_0 = D^{1/2} \mathbb{1} \in \mathbb{R}^n. \tag{2}$$

The eigenvalues $\lambda_i$ — the values on the right-hand side of the problem above — with $0 = \lambda_0 \leq \lambda_1 \leq \cdots \leq \lambda_{n-1} \leq 2$, quantify the normalized smoothness of the functions $y_i$. Since solving the eigenvalue problem has a computational complexity of $O(n^3)$, in practice, we limit the computation to the $k_{max}$ smallest eigenvalues for efficiency. We obtain the spectral gaps $\delta_j = \lambda_j - \lambda_{j-1} > 0$ for $2 \leq j \leq k_{max} - 1$ and seek for the $k$-th gap giving the tighter $\sin \theta$ bound, i.e. $k = \arg \max_j \delta_j$.

We select up to $k$ smoothest normalized functions on $G$, namely the first $k$ eigenvectors of $L$, $y_1, \ldots, y_{k-1}$ — we ignore $y_0$ since it is constant. These $k$ eigenvectors provide orthogonal graph signals based on semantic coherence, with nodes showing similar values in these functions likely belonging to the same semantic part; thus, each signal points to a distinct connected component.

As in [61], we recover the true semantic structure of the graph considering the matrix $Y = [y_1, y_2, \ldots, y_{k-1}] \in \mathbb{R}^{n \times k-1}$. First, we perform $\ell_2$-normalization of each row in $Y$, $X_{ij} = Y_{ij}/(\sum_j Y_{ij}^2)^{1/2} \in \mathbb{R}^{n \times k-1}$ — the $i$-th row of $X$ represents the normalized feature vector for the $i$-th node, which determines the node's membership in a cluster. Then, we take the best membership for each node — using an algorithm that attempts to minimize distortion in a lower-dimensional space — finding $k$ disjoint partitions of the nodes $V_1, V_2, \ldots, V_k$ such that $\bigcup_{i=1}^k V_i = V$ and $\bigcap_{i=1}^k V_i = \emptyset$.

We determine $L'$ and estimate the perturbation $H = L - L'$ to compute the $\sin \theta$ upper bound value.

Each recursion step splits the graph into tighter subgraphs. This process continues until one of the early stopping criteria is met: (1) if a partition is too small (less than $k_{min}$), (2) if the eigenvalues exceed a maximum smoothness threshold $\lambda_{max}$, or (3) if the $\sin \theta$ upper bound value becomes too large (more than $p_{max}$), indicating that the current estimate of $L'$ is no longer reliable. These stopping conditions ensure we halt when further partitioning does not yield meaningful semantic components, thus identifying the final set of image parts or *primitives*.

---

[1]We generalize $G$ and $L$ to denote any subgraph and its Laplacian, rather than just the original image graph.

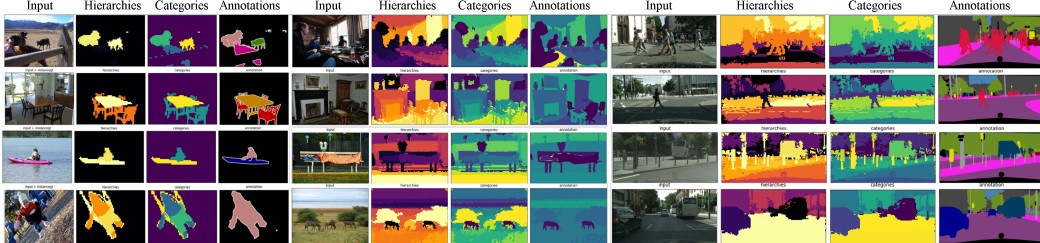

| Input | Hierarchies | Categories | Annotations | Input | Hierarchies | Categories | Annotations | Input | Hierarchies | Categories | Annotations |

Figure 2: Qualitative results of our algorithm on PascalVOC2012, COCO-Stuff and Cityscapes datasets. The *Hierarchy* columns colour-code the pixel semantic hierarchy, and the *Category* columns are random colour-coded, helping visually discriminate hierarchically close pixels.

Each recursive partitioning adds structure to the tree $T$, where nodes specify semantic regions at various levels. The final output is $T$, with each leaf node capturing a distinct part of the image at an appropriate level of granularity; see Figure 2.

The values $k_{min}$, $\lambda_{max}$ and $p_{max}$ are found experimentally for the tested dataset; tables are shown in Section 4 and Appendix D. In Appendix B, we discuss the algorithm's properties and the generated $T$, and present the complete pseudocode of our method.

### 3.2 Pre and Post-Processing

**Boosting computation.** We introduce a preprocessing strategy that condenses the graph, dramatically reducing the algorithm's time and memory requirements by several orders of magnitude while maintaining comparable accuracy. The condensed graph simplifies the original graph into $m$ nodes by contracting strongly connected components into vertices, where $m \ll n$.

We assume that the finest semantic content in natural images is inherently limited and cannot exceed the raw pixel statistics. From the input image $I$ we extract $m$ regions [2, 69, 74] leading to an initial undirected condensed graph $G_c = (V_c, E_c, \tilde{w})$, where $V_c = \{A_i\}_{i=1}^m$, such that $\bigcup_{i=1}^m A_i = V_c$ and $\bigcap_{i=1}^m A_i = \emptyset$, and the edge weights $\tilde{w}(A_i, A_j)$ represent the degree of association between regions, defined as $\tilde{w}(A_i, A_j) = \sum_{u \in A_i, v \in A_j} w(u, v)$. We then apply our recursive partitioning algorithm to $G_c$ and its corresponding normalized Laplacian matrix $L_c$. As a result, we obtain a region tree $T$.

Ablation studies in Section 4.3 compare performances across various overclustering methods.

**Boundary Sharpening.** Given a predicted region tree $T$ with $q$ leaves, $B_1, B_2, \ldots, B_q$, each embedding a disjoint segment of $V$, such that $\bigcup_{j=1}^q B_j = V$ and $\bigcap_{j=1}^q B_j = \emptyset$, we compute the *prototypes* for the image $I$. Each prototype is defined as the $\ell_2$-normalized average of the feature codes in each leaf $u_j = |B_j|^{-1} \sum_{v \in B_j} v$. For each pixel, we define a conditional probability distribution over the prototypes using the softmax function with smoothing parameter $\tau$, namely, $p_{ij} = \exp\left(v_i^\top u_j \tau^{-1}\right) / \sum_k \exp\left(v_i^\top u_k \tau^{-1}\right)$. We arrange the matrix $P = [p_{ij}] \in [0, 1]^{(M \times N) \times q}$ and sharpen the region boundaries applying a Conditional Random Fields (CRF) [49], which refine the predicted distribution $P$ by incorporating dependencies between pixel observations $I$. This improves the accuracy at the boundary, ensuring sharper and more precise segmentation of leaves.

## 4 Experiments

We benchmark our algorithm on unsupervised *multi-granular* segmentation using seven major object- and scene-centric datasets and seven hierarchically structured datasets with varying granularity levels for *hierarchy-agnostic* segmentation. Our evaluation includes an ablation study to assess the contributions of each algorithm component and comparisons across different self-supervised backbone architectures. We only utilize publicly available datasets, SSL model checkpoints without retraining, and validation set ground-truth annotations. CRF is applied only where specified.

Each dataset provides unique characteristics essential for different segmentation challenges. PascalVOC2012 [27] offers a broad range of object categories, making it suitable for general object segmentation tasks. With its high-level division into *things* and *stuff*, COCO-Stuff [10] extends the

MSCOCO [53] and tests the algorithm in complex scenes with multiple objects. Potsdam and Vaihingen [30] datasets, focused on aerial scene parsing, are designed for remote sensing and urban planning applications. Cityscapes [21] is critical for autonomous driving research, providing detailed annotations of urban street scenes. Mapillary Vistats [60] adds diversity with street scenes from various global environments, testing the algorithm's robustness to different conditions. KITTI-STEP [89] and KITTI-SS [1], similar to Cityscapes, extend the evaluation to dynamic urban scenarios with instance detection and object tracking. For fine-grained part segmentation, Pascal-Part [15], PartImageNet, and PartImageNet-158 [35] offer detailed part annotations, crucial for tasks requiring precise recognition and segmentation of object parts.

These datasets ensure a comprehensive and diverse benchmark for evaluating the performance and robustness of our segmentation algorithm across various contexts, see Figure 2. Further details about datasets are in Appendix D.1, and more quantitative and qualitative results are in Appendices D.3 and D.4, respectively.

To ensure reproducibility, we standardize our experimental setup. Unless otherwise specified, we use the `DINOv2-ViT-B14-REG` [22] backbone with parameters $k_{min} = 1$, $p_{max} = 20$, and $\lambda_{max} = 0.8$. We apply the spectral method from Ng et al. [61] with $m = 300$ for superpixel clustering. The recursive partitioning depth is limited at 10 levels. Depending on each backbone downsampling factor, input images are resized to extract $60 \times 60$ codes, except for urban street scenes, where we obtain $60 \times 120$ codes. Further details in Appendix D.

## 4.1 Evaluation Metrics

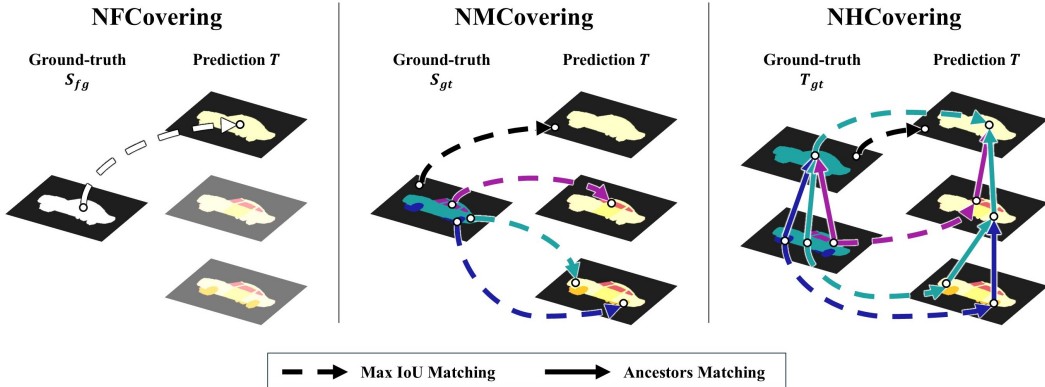

Figure 3: **Comparison of segmentation metrics.** NFCovering evaluates single-level foreground overlap, NMCovering extends across multiple granular levels for all categories, and NHCovering integrates hierarchical consistency. Coloured arrows indicate category-specific matches.

**Granularity-agnostic.** Following [44], we aim to evaluate the unbiased overlapping of regions between predicted segmentation and ground truth within each image via the *Normalized Foreground Covering* (NFCovering) metric. However, it is not well-suited for the multi-granular segmentation domain. The metric applies to a single granularity level at a time, failing to account for multiple granularity levels. Furthermore, it disregards the background as a valid semantic instance, leading to an incomplete estimate of segmentation performance.

We propose a novel evaluation metric, the *Normalized Multigranular Covering* (NMCovering), which addresses these limitations evaluating the overlap of regions between the unrolled segments in the region tree $T$ and all the available ground-truth categorical segments in a semantic map $S_{gt}$. This metric ensures that both foreground and background instances are considered, providing a more comprehensive and granularity-independent assessment of a semantic segmentation model's performance. We adopt a greedy heuristic that maximises the overlap of the full hierarchy with the ground truth segmentation and compute the average overlap ratio of the matching:

$$\text{NMCovering}(T \rightarrow S_{gt}) := \frac{1}{|S_{gt}|} \sum_{R \in S_{gt}} \max_{R' \in T} \frac{|R \cap R'|}{|R \cup R'|}. \tag{3}$$

Table 1: **Granularity-agnostic.** Evaluation of our algorithm on different datasets using a maximum overlap heuristic for category matching.

| Dataset | mIoU | pAcc | mAcc | fIoU | NMCovering $(T \rightarrow S_{gt})$ |
|---|---|---|---|---|---|
| *object-centric* | | | | | |
| PascalVOC2012 | 78.1 | 82.6 | 91.2 | 78.1 | 75.4 |
| MSCOCO | 55.7 | 93.1 | 85.0 | 78.8 | 49.6 |
| *scene-centric* | | | | | |
| COCO-Stuff | 58.7 | 81.1 | 80.3 | 67.3 | 42.1 |
| Cityscapes | 48.8 | 82.8 | 76.1 | 68.8 | 44.8 |
| KITTI-STEP | 51.2 | 79.8 | 76.5 | 65.7 | 48.4 |
| Mapillary Vistas | 47.6 | 78.9 | 72.1 | 66.1 | 42.7 |
| Potsdam | 58.9 | 83.4 | 83.2 | 65.0 | 56.3 |

Table 2: **Hierarchy-agnostic.** Evaluation of our algorithm on different datasets using a maximum overlap heuristic for category matching.

| Dataset | mIoU | pAcc | NMCovering $(T \rightarrow T_{gt})$ | NHCovering |
|---|---|---|---|---|
| *whole-centric* | | | | |
| COCO-Stuff | 59.5 | 75.1 | 53.5 | 42.9 |
| Cityscapes | 53.7 | 78.8 | 51.1 | 43.8 |
| KITTI-STEP | 58.3 | 79.6 | 54.2 | 46.5 |
| Mapillary Vistas | | | | |
| *part-centric* | | | | |
| Pascal-Part | 25.8 | 80.0 | 39.5 | 38.8 |
| Part-Imagenet | 55.4 | 79.5 | 65.8 | 65.2 |
| Part-Imagenet-158 | 59.5 | 82.6 | 67.8 | 63.1 |

The NMCovering metric evaluates the performance of a hierarchical segmentation model considering how many ground truth objects are recognised at any granularity level and how well they are segmented. A high score indicates a high segmentation coherence between human semantic perception and unsupervised machine one.

**Hierarchy-agnostic.** We introduce a second accuracy metric, the *Normalized Hierarchical Covering* (NHCovering). NHCovering jointly evaluates the segmentation quality and the semantic *hierarchical inclusion* of the prediction relating to the ground-truth semantic region tree $T_{gt}$. Hierarchical inclusion is the matching between the nodes of two distinct hierarchies preserving the lineage from the ancestors; this problem is commonly referred to in the literature as the *unordered tree inclusion problem* [45].

To calculate this metric, we use a greedy heuristic that computes the average overlap ratio of matching regions, weighting each match by the ratio of matched ancestors. The operator $\pi(R)$ returns the ancestors set of the tree node $R$, and $\beta(R, T)$ returns the nodes set in the predicted tree $T$ that best match the ancestors of node $R$:

$$\text{NHCovering}(T \rightarrow T_{gt}) := \frac{1}{|T_{gt}|} \sum_{R \in T_{gt}} \max_{R' \in T} \frac{|R \cap R'|}{|R \cup R'|} \cdot \frac{|\beta(R, T) \cap \pi(R')|}{|\pi(R)|}, \qquad (4)$$

$$\text{where } \beta(R, T) := \bigcup_{P \in \pi(R)} \arg\max_{P' \in T} \frac{|P \cap P'|}{|P \cup P'|}. \qquad (5)$$

The NHCovering metric computes the average weighted overlap of regions between the predicted tree $T$ and the ground-truth tree $T_{gt}$. The overlap weight measures the proportion of correct ancestorships with respect to the ground truth. Balancing segmentation performance with semantic ancestry consistency provides a granularity- and hierarchy-independent performance assessment. This score quantifies the coherence of segmentation and hierarchical organization of visual concepts between human perception [66] and unsupervised machine one.

Refer to Figure 3 for a visual insight into the metrics. A more detailed discussion is in Appendix D.2.

## 4.2 Unsupervised Segmentation

**Granularity-Agnostic.** We adopt the NMCovering metric to benchmark the performance and versatility of our algorithm across different natural image domains. As shown in Tables 1 and 5, our method achieves excellent results in segmenting object-centric images and foreground discovery. Additionally, Table 1 demonstrates our approach's strong performance on scene-centric datasets, such as remote-sensing images and urban street scenes. Table 3 compares our approach to other supervision strategies.[2] Our approach achieves segmentation quality comparable to supervised methods and surpasses other supervision strategies by a large gap.

**Hierarchy-Agnostic.** We further benchmark the hierarchical inclusion quality of the algorithm on available datasets having hierarchical structures at a high level, such as MSCOCO, COCO-Stuff and Cityscapes, and at a low level, such as PascalPart and PartImageNet. We show in Table 2

---

[2]We ran four experiments for each dataset with random seed and assumed normally distributed errors. However, in the segmentation literature, mIoU is typically reported by the single mean value in percentage.

Table 3: **Semantic segmentation.** Comparison on PascalVOC2012 *val*. Ours match unsupervised masks to best overlapping classes.

| Method | Backbone | mIoU VOC12 | mIoU MSCOCO |
|---|---|---|---|
| *fully-supervised* | | | |
| DeepLab-CRF [12] | ResNet-101 | 77.7 | - |
| DeepLab-CRF [12] | VGG-16 | - | **43.6** [10] |
| DeepLabV3-JFT [13] | ResNet-101 | **82.7** | - |
| *weakly-supervised* | | | |
| ViT-PCM [71] | ViT-B16 | 69.3 | 45.0 |
| L2G [42] | ResNet-38 | 72.0 | 44.2 |
| WeakTr [95] | DeiT-S | **74.0** | **50.3** |
| *un-supervised* | | | |
| Melas-Kyriazi et al. [58] | ViT-S16 | 37.2 | - |
| Leopart [96] | ViT-S16 | 41.7 | 49.2 |
| HSG [44] | ResNet-50 | 41.9 | - |
| Zhang et al. [94] | ResNet-50 | 43.5 | - |
| MaskDistill [79] | ResNet-50 | 48.9 | - |
| **Ours w/o CRF** | ViT-S8 | 76.2 ± .9 | 52.1 ± .6 |
| **Ours w CRF** | ViT-B14 | **80.3 ± 1.1** | **56.5 ± .9** |

Table 4: **Boundary potential methods.** All methods match unsupervised tree segments to best overlapping classes.

| PascalVOC2012 | mIoU | pAcc | NMCovering $(T \rightarrow S_{gt})$ |
|---|---|---|---|
| *boundary potential* | | | |
| SE-OWT-UCM [24] | 48.4 | 83.0 | 59.0 |
| PMI-OWT-UCM [40] | 47.0 | 86.5 | 61.3 |
| *semantic smoothness* | | | |
| **Ours w/o CRF** | 78.1 | 86.0 | 75.4 |
| **Ours w CRF** | **80.3** | **87.3** | **76.8** |

| COCO-Stuff | mIoU | NMCovering $(T \rightarrow T_{gt})$ | NHCovering |
|---|---|---|---|
| *boundary potential* | | | |
| SE-OWT-UCM [24] | 30.7 | 43.0 | 32.9 |
| PMI-OWT-UCM [40] | 27.5 | 43.2 | 23.1 |
| *semantic smoothness* | | | |
| **Ours w/o CRF** | 58.7 | 53.5 | 42.1 |
| **Ours w CRF** | **59.9** | **55.6** | **43.9** |

Table 5: **Backbone ablation.** Granularity-agnostic segmentation evaluation on PascalVOC2012 *val* set using a maximum overlap heuristic for category matching in each image. We report category IoU for each experiment with micro and macro averaged scores and the NMCovering.

| Backbone | bkgd | airplane | bicycle | bird | boat | bottle | bus | car | cat | chair | cow | d. table | dog | horse | bike | person | p. plant | sheep | couch | train | tv | mIoU | pAcc | mAcc | fIoU | NMCovering |
|---|---|---|---|---|---|---|---|---|---|---|---|---|---|---|---|---|---|---|---|---|---|---|---|---|---|---|
| ViT-B8 [25] | 63.9 | 58.5 | 40.1 | 60.5 | 58.0 | 59.7 | 74.1 | 68.6 | 68.8 | 49.7 | 67.5 | 52.0 | 65.6 | 68.6 | 58.5 | 60.5 | 58.1 | 66.5 | 62.4 | 64.2 | 52.4 | 60.9 | 69.8 | 75.1 | 63.6 | 60.8 |
| CLIP-ViT-B16 [67] | 74.4 | 73.0 | 52.2 | 82.0 | 71.2 | 66.5 | 76.5 | 84.0 | 87.4 | 66.4 | 86.3 | 59.1 | 83.2 | 80.3 | 75.0 | 76.1 | 70.0 | 85.5 | 79.2 | 70.5 | 63.3 | 74.4 | 79.5 | 84.0 | 75.1 | 74.0 |
| MAE-ViT-B16 [37] | 66.2 | 81.4 | 54.6 | 85.8 | 73.4 | 71.4 | 82.4 | 80.6 | 83.8 | 64.8 | 85.1 | 66.8 | 83.8 | 81.4 | 74.6 | 72.6 | 66.5 | 87.3 | 77.0 | 76.2 | 68.7 | 75.4 | 73.5 | 85.9 | 69.1 | 70.0 |
| MOCOv3-ViT-B16 [16] | 72.2 | 82.6 | 57.2 | 83.0 | 74.4 | 69.9 | 78.7 | 76.1 | 81.8 | 59.0 | 85.7 | 66.7 | 80.3 | 77.2 | 72.3 | 70.6 | 60.2 | 86.4 | 77.6 | 76.4 | 61.8 | 73.8 | 78.1 | 84.9 | 73.0 | 71.1 |
| DINO-ResNet-50 [11] | 67.2 | 65.7 | 47.6 | 70.2 | 58.8 | 49.8 | 66.8 | 56.6 | 73.9 | 46.8 | 73.6 | 47.1 | 70.3 | 71.6 | 60.7 | 55.2 | 52.6 | 77.5 | 59.5 | 63.7 | 39.2 | 60.8 | 73.3 | 76.0 | 65.7 | 61.9 |
| DINO-ViT-S8 [11] | 69.7 | 83.1 | 51.7 | 85.8 | 75.2 | 70.2 | 84.0 | **82.0** | 86.7 | 67.1 | 85.8 | 66.3 | 85.8 | 80.0 | 76.5 | 73.5 | 66.3 | 86.4 | 81.3 | 75.9 | 66.9 | 76.2 | 76.8 | 85.6 | 72.0 | 72.5 |
| DINO-ViT-B8 [11] | 70.6 | **87.0** | 57.1 | **91.0** | **77.1** | 74.3 | 83.7 | 80.0 | 88.1 | 67.5 | 86.2 | 65.2 | 85.5 | **81.2** | 78.6 | 75.0 | 66.2 | **88.9** | **83.5** | 80.0 | 67.6 | 77.8 | 77.4 | 86.0 | 73.0 | 74.0 |
| DINOv2-ViT-B14-R [22] | **76.9** | 73.4 | 51.0 | 82.1 | 72.4 | **82.5** | **85.6** | 81.1 | **90.2** | **71.2** | **87.1** | **68.8** | **87.7** | 78.3 | **79.2** | **82.1** | **70.8** | 84.7 | 82.9 | **82.9** | **68.8** | **78.1** | **82.6** | **91.2** | **78.1** | **75.4** |

the closeness in performance of NHCovering with respect to the NMCovering, demonstrating the ability of the algorithm to capture hierarchical relations among the parts. The lower performance on PascalPart is due to the lower granularity of part annotations compared to PartImageNet, see [35].

Recent unsupervised semantic segmentation approaches, such as [44, 94], often employ mutual information maximisation between regions at multiple granularity levels. These methods typically utilise hierarchical clustering that groups low-level coherent regions via boundary potentials, such as the Ultrametric Contour Map (OWT-UCM) [4], of boundaries derived from low-level features like brightness, colour, and texture gradients, as in Structured Edges (SE) [24] or Pointwise Mutual Information (PMI) [40]. We compare our approach with these methods in Table 4. The results indicate that low-level processes are inappropriate for the hierarchical grouping of high-level (semantic) features. In contrast, our approach excels in this area, suggesting significant room for improvement in the current state of the art. See some hierarchical grouping results in Figure 4.

## 4.3 Ablation Experiments

**Backbone.** We evaluate in Table 5 the consistency of our approach according to the latent space induced by different SSL Imagenet pre-trained backbones on the granularity-agnostic task on the

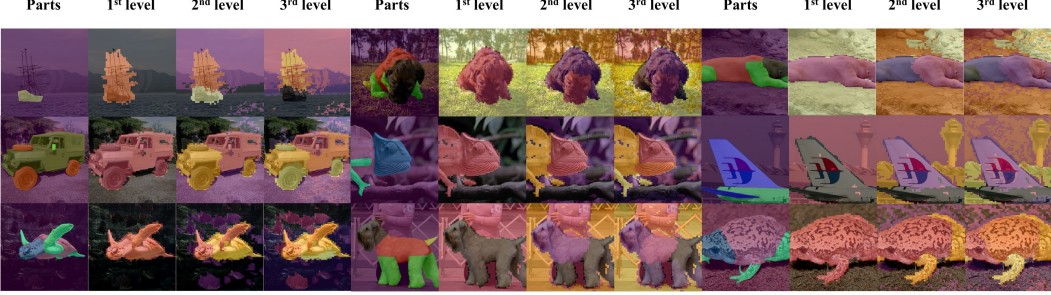

Figure 4: **Unsupervised parts discovering examples on PartImageNet.** The left column shows the ground truth part masks. The second to fourth column shows the predicted regions for each tree depth. Heatmap colours encode leaves' distance in the subtrees.

Table 6: **Superpixel and parameters ablation experiments.**

(a) NMCovering on PartImageNet: superpixel vs. $k_{min}$.

| Superpixel $m = 100$ | $k_{min}$ 1 | 5 | $k_{min}=1$ (sec/iter) |
|---|---|---|---|
| *colour-space* | | | |
| k-mean [54] | 32.7 | 33.4 | .73 ± .14 |
| SLIC [2] | 58.1 | 44.9 | **.05** ± .01 |
| quick-shift [80] | **60.8** | 58.0 | .21 ± .12 |
| *SSL-latent-space* | | | |
| k-mean [54] | 62.9 | 58.1 | **.34** ± .19 |
| Spectral [61] | **63.1** | **59.9** | .61 ± .11 |
| *None* | 65.7 | 64.9 | .84 ± .29 |

(b) mIoU for $m$ sizes with [61].

| Dataset | $m$ 50 | 100 | 300 | None |
|---|---|---|---|---|
| *object-centric* | | | | |
| PascalVOC2012 | 76.3 | 77.5 | 78.1 | 78.1 |
| MSCOCO | 45.8 | 52.4 | 55.7 | 56.1 |
| *scene-centric* | | | | |
| COCO-Stuff | 43.6 | 52.2 | 59.5 | 60.3 |
| Cityscapes | 30.3 | 37.6 | 48.8 | 50.4 |
| *part-centric* | | | | |
| Pascal-Part | 18.2 | 23.1 | 25.8 | 26.4 |
| PartImageNet | 36.9 | 47.2 | 48.3 | 48.8 |

(c) NHCovering with different perturbation thresholds and smoothness parameters on the COCO-Stuff dataset.

| $p_{max}$ | $\lambda_{max}$ .5 | .8 | None | $\lambda_{max}=.5$ (sec/iter) |
|---|---|---|---|---|
| 15 | 36.9 | 41.1 | 41.4 | .49 ± .09 |
| 20 | 39.5 | 42.9 | 43.1 | .65 ± .16 |
| *None* | 40.9 | 43.7 | 44.0 | .77 ± .23 |

PascalVOC2012 *val* set. The performance gap reflects the representation quality assessed by SSL downstream task benchmarks. Such a result assesses the best model and a complementary downstream task benchmark for SSL. We do not adopt superpixel clustering or CRF but utilise raw patch features as pixel codes.

**Parameters $k_{min}$, $p_{max}$ and $\lambda_{max}$.** In Tables 6a to 6c we validate the optimal parameters of our algorithm. While $k_{min}$ choice affects the granularity at lower levels, the $p_{max}$ and $\lambda_{max}$ choice affects the granularity at higher levels by controlling the stability of the partition.

**Overclustering and CRF.** We test different over clustering techniques in Table 6a. Results show higher performances for a simultaneous normalised cut on SSL latent space. When applying CRF with $\tau = 0.1$, we obtain an increase in segmentation accuracy as shown in Tables 3 and 4.

## 5 Discussion

**Broader impact.** Supervised learning typically relies on highly curated datasets that require expensive and time-consuming manual annotations, especially for complex tasks that demand expert knowledge, such as fine-grained semantic segmentation. As the demand for larger datasets grows and costs rise, increasing interest is in advancing image understanding with minimal or no supervision. Our approach addresses this challenge by streamlining the annotation process, reducing costs, and thereby significantly expanding the data available for supervised model training through the dynamic discovery of semantic regions.

**Limitations.** While our method shows promising results, it does have limitations. One major drawback is that both segmentation quality and algorithm execution time scale with the input size. Small object parts are especially difficult to detect, particularly when overclustering is used during preprocessing; the results support this. However, a trade-off between accuracy and inference time can be experimentally determined. Moreover, our approach is based on self-supervised learning, which, like all data-driven methods, is susceptible to inherent biases in the data. These biases can influence the resulting latent space of the model, potentially impacting the performance and generalization of our approach.

## 6 Conclusions

We introduce a novel method for unsupervised hierarchical decomposition of natural scenes into primitive components, without requiring prior knowledge of scene granularity. Leveraging deep feature extraction and graph partitioning, our approach constructs a tree of semantic elements from any scene for any dataset. Our core algorithm applies an innovative algebraic approach to deep spectral clustering, addressing blurring from pixel similarity across object parts. Matrix perturbation theory is employed at each tree level, ensuring stable smooth partitions. This framework not only advances unsupervised semantic segmentation but also benchmarks deep neural network representations by evaluating segmentation quality at multiple granularity levels and hierarchical consistency among them. We validate our method with novel metrics, evaluating overlap with ground-truth masks across diverse semantic segmentation datasets and semantic inclusion within hierarchical ones.

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

# A    Theoretical Analysis and Perturbation Foundations

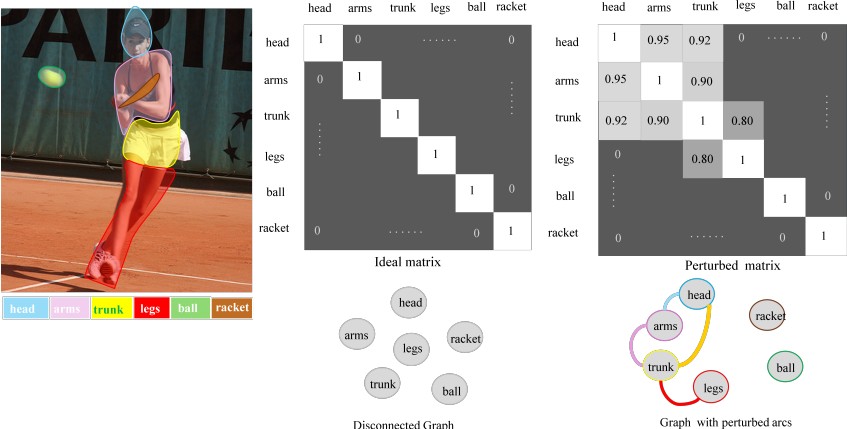

Figure 5: **An example of ideal and perturbed adjacency matrices.** The left shows an input image with highlighted parts and a colour legend. The central matrix represents the ideal adjacency matrix $W'$, corresponding to the Laplacian $L'$, with non-zero diagonal blocks for $k'$ disconnected components at a specific semantic granularity. Below, a disconnected graph illustrates these isolated parts. On the right, the perturbed adjacency matrix $W$ introduces off-diagonal entries due to pixel similarity across regions, resulting in the perturbed Laplacian $L$. Below, a graph with added connections shows these perturbations, with colours matching the highlighted parts in the input image.

From the example in Figure 5, we observe that perturbation arises from pixel similarity values that inadvertently create connections between regions, which ideally should remain separate. This unintended overlap occurs because similarity values between certain pixel pairs do not perfectly align with the true structure of the components. Perturbation theory [23, 9, 75, 43] studies how small changes, such as these unintended connections, affect a matrix's eigenspaces. Recent research has expanded this analysis from Hermitian matrices to include Laplacian matrices, which capture graph structures, as in [82, 26, 70, 28, 65, 91].

In our method, the two matrices are an ideal Laplacian matrix $L'$ and the observed Laplacian matrix $L$, both symmetric (not necessarily of the same rank), perturbed by a symmetric matrix $H$ induced by higher-order pixel similarity, $L = L' + H$. Given the recursive partitioning described in Section 3, we are looking, for each subgraph, for the best matches between the eigenvectors of $L'$ and those of $L$ minimising the functional defined in Equation (1).

In the ideal normalized Laplacian $L'$, the graph is disconnected, and the number of connected components is reflected in the *multiplicity of the zero eigenvalue* of $L'$. Each connected component contributes one zero eigenvalue, and the *constant* eigenvectors corresponding to these zero eigenvalues represent these isolated components [20, 81]. In a perturbed normalized Laplacian $L$, the graph is connected and then the minimum of Equation (2) is achieved by the eigenvector corresponding to the second smallest eigenvalue of $L'$. This eigenvector, known as the Fiedler vector, is *not constant* and captures the connectivity structure of the graph, often used for finding the best bipartition.

However, in the case of a perturbed graph with weak connections (e.g., low weights or nodes with only a single neighbour), small eigenvalues close to zero may not indicate strongly connected subgraphs. Instead, these small eigenvalues reflect loosely connected regions, where perturbations create weak links between components that ideally should remain separate. This implies that using the smallest eigenvectors directly for graph partitioning is sometimes unreliable, as they may reflect unstable or artificial connections introduced by perturbations. To ensure meaningful segmentation, it is essential to establish a measure that quantifies the stability of the partition under perturbations. Such a measure would indicate when it is safe to apply spectral clustering, ensuring that the identified clusters are robust and well-separated. According to Theorem A.1 (shown below), to achieve this, we seek the set of eigenvectors of the perturbed Laplacian $L$ that are closest to the eigenvectors of the ideal normalized Laplacian $L'$ despite the perturbation $H$:

**Theorem A.1** ([91]). *Let $A, A' \in \mathbb{R}^{n \times n}$ be symmetric, with eigenvalues $\mu_1 \geq \cdots \geq \mu_n$ and $\mu'_1 \geq \cdots \geq \mu'_n$ respectively. Fix $1 \leq r \leq s \leq n$ and assume that $\min(\mu_{r-1} - \mu_r, \mu_s - \mu_{s+1}) > 0$, where $\mu_0 := \infty$ and $\mu_{n+1} := -\infty$. Let $u := s - r + 1$, and let $Z = (z_r, z_{r+1}, \ldots, z_s) \in \mathbb{R}^{n \times u}$ and $Z' = (z'_r, z'_{r+1}, \ldots, z'_s) \in \mathbb{R}^{n \times u}$ have orthonormal columns satisfying $A z_j = \mu_j z_j$ and $A' z'_j = \mu'_j z'_j$, for $j = r, r+1, \ldots, s$. Then:*

$$\| \sin \Theta(Z', Z) \|_F \leq \frac{2 \min(u^{1/2} \| A' - A \|_{op}, \| A' - A \|_F)}{\min(\mu_{r-1} - \mu_r, \mu_s - \mu_{s+1})} \tag{6}$$

*Moreover there exists and orthogonal matrix $\mathcal{O}'$ in $\mathbb{R}^{n \times n}$ such that:*

$$\| Z' \mathcal{O}' - Z \|_F \leq \frac{2^{3/2} \min(u^{1/2} \| A' - A \|_{op}, \| A' - A \|_F)}{\min(\mu_{r-1} - \mu_r, \mu_s - \mu_{s+1})} \tag{7}$$

With $op$ and $F$, the spectral and the Frobenius norm, respectively.

**Corollary A.1** ( [91]). *Let $A, A' \in \mathbb{R}^{n \times n}$ be symmetric, with eigenvalues $\mu_1 \geq \cdots \geq \mu_n$ and $\mu'_1 \geq \cdots \geq \mu'_n$ respectively. Fix $j \in \{1, \ldots, n\}$ and assume that $\min(\mu_{j-1} - \mu_j, \mu_j - \mu_{j+1}) > 0$, where $\mu_0 := \infty$ and $\mu_{n+1} := -\infty$. If $z, z' \in \mathbb{R}^n$ satisfy $A z = \mu_j z$ and $A' z' = \mu'_j z'$, then:*

$$\| \sin \Theta(z', z) \| \leq \frac{2 \| A' - A \|_{op}}{\min(\mu_{j-1} - \mu_j, \mu_j - \mu_{j+1})} \tag{8}$$

*Moreover, if $z'^\top z \geq 0$, then:*

$$\| z' - z \| \leq \frac{2^{3/2} \| A' - A \|_{op}}{\min(\mu_{j-1} - \mu_j, \mu_j - \mu_{j+1})} \tag{9}$$

In our method, the above symmetric matrix $A$ refers to the perturbed matrix $L$ and the matrix $A'$ to the ideal matrix $L'$; see Section 3. Since we select the $k$ smallest eigenvalues, the interval starts from $r = n - k + 1$ and ends at $s = n$, and we have $u = k$. The denominator in Equation (6) simplifies to $\min(\mu_{n-k} - \mu_{n-k+1}, \mu_n - \mu_{n+1}) = \mu_{n-k} - \mu_{n-k+1}$, since by definition $\mu_{n+1} := -\infty$, and, given $Z = (z_{n-k+1}, z_{n-k+2}, \ldots, z_n) \in \mathbb{R}^{n \times k}$ and $Z' = (z'_{n-k+1}, z'_{n-k+2}, \ldots, z'_n) \in \mathbb{R}^{n \times k}$, depending on the chosen eigenvalue ordering, we have:

$$\| \sin \Theta(Z', Z) \|_F \leq \frac{2 \min(k^{1/2} \| L' - L \|_{op}, \| L' - L \|_F)}{\mu_{n-k} - \mu_{n-k+1}} \quad \text{for} \quad \mu_1 \geq \cdots \geq \mu_n, \tag{10}$$

or, reversing the indexing — eigenvalues in non-decreasing order — and counting from zero, we set $\lambda_i = \mu_{n-i}$ and $\lambda'_i = \mu'_{n-i}$ for $i = 0, \ldots, n-1$ with $Y = (y_0, y_1, \ldots, y_{k-1}) \in \mathbb{R}^{n \times k}$ and $Y' = (y'_0, y'_1, \ldots, y'_{k-1}) \in \mathbb{R}^{n \times k}$ the orthonormal columns satisfying $L y_j = \lambda_j y_j$ and $L' y'_j = \lambda'_j y'_j$, for $j = 0, 1, \ldots, k-1$, we have:

$$\| \sin \Theta(Y', Y) \|_F \leq \frac{2 \min(k^{1/2} \| L' - L \|_{op}, \| L' - L \|_F)}{\lambda_k - \lambda_{k-1}} \quad \text{for} \quad \lambda_0 \leq \cdots \leq \lambda_{n-1}. \tag{11}$$

Indeed, we can apply the substitution and the indexing, given the eigenpairs $(\lambda, y)$ condition for all the eigenvalues and eigenvectors populating the chosen interval $(r, s)$. Furthermore, given the eigenpair condition, the minimization of Equation (1) in Section 3 is guaranteed (see Algorithm 1), and we can just ratify the smoothness by resorting to the Courant-Fisher theorem; see [38] Theorem 4.2.6. Finally, we can use the Corollary A.1 for a more refined choice of the eigenvectors.

Note that the theorem of Yu et al. [91] is particularly useful because, differently from Davis-Kahan [23] theorem, it defines an upper bound between two symmetric matrices concerning the angles between a subset of the eigenvectors of the two matrices or their distance (up to a rotation), in terms of the eigenvalues of one of the two matrices. Indeed, this theorem shows that $z$ (an eigenvector of the perturbed Laplacian $L$) is close to $z'$ (an eigenvector of the Laplacian $L'$), under two main assumptions. First, we assume that $L$ is close to $L'$ — often this is straightforward in graph theory, for instance, if $L'$ is derived from a theoretical (or "population") graph structure, and $L$ is the Laplacian of a graph constructed from a sample or noisy measurements of this structure. Second, applying Weyl's inequality, we assume that, almost certainly:

$$|\mu'_{j-1} - \mu_j| \geq (\mu_{j-1} - \mu_j)/2 \quad \text{and} \quad |\mu'_{j+1} - \mu_j| \geq (\mu_j - \mu_{j+1})/2,$$

where $\mu_j$ and $\mu_j'$ are the $j$-th eigenvalues of $L$ and $L'$, respectively. Under this eigengap condition, assuming a sufficient separation between the population eigenvalues of $L$ and their neighbouring values, as shown in Yu et al. [91] results, we can conclude that $\|z' - z\|$ is small, meaning $z$ and $z'$ are close in norm. Building on this, the theorem implies that when the eigengap condition holds, the eigenvector $z$ associated with $L$ remains stable under perturbations represented by $H = L - L'$. Specifically, the proximity of $z$ and $z'$ allows us to interpret $z$ as a meaningful approximation of $z'$, preserving the structure of the ideal graph encoded by $L'$.

Consequently, this alignment of eigenvectors facilitates robust graph-based clustering or segmentation, as it enables us to consistently identify clusters or partitions in perturbed graphs that mirror the structure of the unperturbed graph. Furthermore, this result provides a foundation for using the perturbed eigenvectors for hierarchical clustering by ensuring that the segments or clusters derived from $L$ approximate those of $L'$ even under small changes or noise in the graph data.

## B  The Algorithm and BFS Implementation

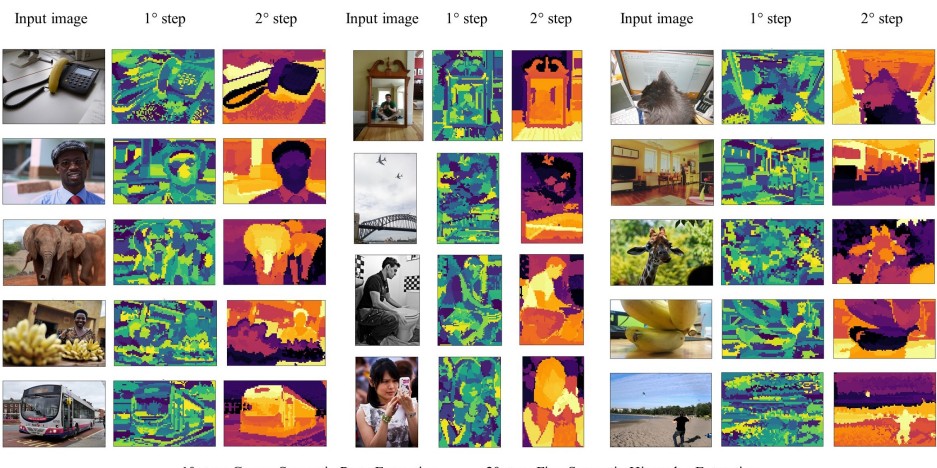

1° step: Coarse Semantic Parts Extraction        2° step: Fine Semantic Hierarchy Extraction

Figure 6: **The algorithm's two steps outputs.** First, we quantize the graph to create an initial over-clustering of semantic parts. Next, we recursively group these parts, forming multi-level semantic clusters from coarse to fine granularity. The heatmap colour-codes the distance between tree leaves.

In Section 3 of the main paper, we have presented the recursive partitioning of the graph $G$, which is simple and intuitive. As noted, we present a Breadth-First Search (BFS) pseudocode in Algorithm 1.

The method obtains graph partitions when two or more components are detectable and recur on each partition to find new subgraphs until an early stopping condition is met, see Section 3. We report that the proposed algorithm is defined in two steps for completeness — the first step is meant to speed up the overall algorithm, and hence, it is optional. In the first step, we quantise the graph, creating an over-clustering of semantic parts; in the second step, we recur on the parts, generating coarse to finer semantic groups at multiple granularity levels; see Figure 6.

In the first step, we write the patch-wise features extracted with a deep neural network as the set of nodes $V$ of a weighted undirected graph $G = (V, E, w)$, and we define $w$ as the cosine similarity of points, scaled and shifted in $[0, 1]$.

In this step, we follow the approach of Ng et al. [61] to cluster the graph in $m$ components. The clusters define $m$ disjoint segments $A_1, A_2, \ldots, A_m$ ($\bigcup_{i=1}^{m} A_i = V$, $\bigcap_{i=1}^{m} A_i = \emptyset$) of $G$ with high intra-cluster degree of semantic similarity.

In the second step, we adopt a top-down *recursive* divisive clustering approach, building a hierarchical semantic decomposition of the image. In the recursive call at a certain level $l$ and for a certain subgraph $c$ — notice that here $c$ is an index — of a graph $p$ we define an undirected condensed graph $G_c^{l,p} = (V_c^{l,p}, E_c^{l,p}, \tilde{w})$, with segments $A$ as nodes in $V_c^{l,p}$ ($\bigcup_{A \in V_c^{l,p}} A \subseteq V$, $\bigcap_{A \in V_c^{l,p}} A = \emptyset$) and the edges are weighted by the associativity degree of the components

$\tilde{w}(A_i, A_j) = \sum_{u \in A_i, v \in A_j} w(u, v)$. Notice that the condensed graph at the root level is indicated as $G_0^{0,0} = (V_0^{0,0}, E_0^{0,0}, \tilde{w})$ and $V_0^{0,0} = \{A_i\}_{i=1}^m$. At each depth, we obtain a semantic grouping $S^l = \{V_i^{l+1,c}\}_{c,i}$ given by all the $i$-th subgraphs at level $l + 1$ of each subgraph $c$ at level $l$.

The recursion defines the final hierarchy, namely the output tree $T$.

The semantic grouping algorithm discussed here builds upon conventional spectral clustering methods [74, 61, 6, 81].

---

**Algorithm 1:** Image Parsing via Granularity and Hierarchy-agnostic Semantic Regions Tree

---

**Data:** $V$: set of points
         $w$: points similarity function
         $m$: desired number of superpixels
         $\tilde{w}$: components similarity function
         $k_{max}$: max number of subgraph components
         $k_{min}$: min number of subgraph points
         $\lambda_{max}$: max normalized smoothness threshold
         $p_{max}$: max perturbation threshold

**Result:** $\ell$-depth hierarchy of clusterings $\{S^l\}_{l=1}^\ell$

$V_0^{0,0} = \{A_i\}_{i=1}^m = \texttt{spectral\_overclustering}(V)$
$\{S^l\}_{l=1}^\ell = \texttt{bfs\_partitioning}(\{V_0^{0,0}\})$

**Function** $\texttt{spectral\_overclustering}(V)$:
    Compute $L$ for graph $G = (V, E, w)$
    $Y \leftarrow \arg\min_{Y \in \mathbb{R}^{|V| \times m}, Y^\top Y = I} \text{Tr}(Y^\top L Y)$
    Normalize rows $X_{ij} = Y_{ij}/(\sum_j Y_{ij}^2)^{1/2}$
    Group $X$ rows in $m$ clusters with K-means
    **return** $m$ groups $\{A_i\}_{i=1}^m$ of $V$

**Function** $\texttt{bfs\_partitioning}(S^l)$:
    $S^{l+1} \leftarrow \{\}$
    **foreach** *segment* $V_c^{l,p} \in S^l$ **do**
       $n \leftarrow |V_c^{l,p}|$
       **if** $n < k_{min}$ **then**
          **continue**
       **end**
       Get condensed graph $G \leftarrow G_c^{l,p} = (V_c^{l,p}, E_c^{l,p}, \tilde{w})$
       Scale $W$ with min-max normalization and compute $L$
       $Y \leftarrow \arg\min_{Y \in \mathbb{R}^{n \times k_{max}}, Y^\top Y = I} \text{Tr}(Y^\top L Y)$
       $\lambda \leftarrow \text{diag}(Y^\top L Y)$
       Reorder $Y$ columns and $\lambda$ in ascending order of $\lambda_i$
       $k \leftarrow \arg\max_i \{\lambda_i - \lambda_{i-1}\}_{i=2}^{k_{max}-1}, \lambda_{i-1} < \lambda_{max}$
       **if** $\nexists k$ **then**
          **continue**
       **end**
       Take first $k$ eigenvectors $Y \leftarrow Y_{[:,1:k]}$
       Normalize rows $X_{i,j} = Y_{i,j}/(\sum_j Y_{i,j}^2)^{1/2}$
       Group $X$ rows in $k$ clusters with K-means and get $L'$
       Compute perturbation $H = L - L'$
       **if** $2 \min(k^{1/2} \|H\|_{op}, \|H\|_F)/(\lambda_k - \lambda_{k-1}) > k \, p_{max}$ **then**
          **continue**
       **end**
       Found stable $k$ subgraphs estimate $S_c^{l+1} = \{V_i^{l+1,c}\}_{i=1}^k$ of $V_c^{l,p}$
       $S^{l+1} \leftarrow S^{l+1} \cup S_c^{l+1}$
    **end**
    **return** $\{S^{l+1}\} \cup \texttt{bfs\_partitioning}(S^{l+1})$

---

**Discussion above the algorithm.** The algorithm's design integrates several properties that significantly shape semantic components' hierarchical structure and connectivity.

One crucial feature is the dynamic estimation of the number of components, $k$, a parameter closely aligned with insights from perturbation theory. This estimate directly influences segmentation granularity, determining the expected number of distinct semantic regions and hierarchy levels in the final output.

The algorithm produces a hierarchy-agnostic unsupervised semantic region tree $T$. At each recursion stage, it seeks subgraphs that represent loosely connected regions based on robust algebraic criteria from graph theory, capturing coherent semantic regions at each level of granularity. By isolating each region from the context of previous layers, the algorithm accurately reflects the semantic hierarchy within the image's content. In this framework, the tree's leaves represent regions with high intra-cluster semantic similarity, positioning these clusters as primitives of broader concepts at higher levels. The algorithm naturally embeds semantic inclusion, where finer semantic regions exist within larger semantic contexts, mirroring how complex concepts encompass finer details.

Lastly, computing the adjacency matrix only once optimizes efficiency, reducing redundant operations and improving runtime performance, making the approach both scalable and effective for large datasets.

**Time Complexity.** The breadth-first search has a time complexity of $\mathcal{O}(|V| + |E|)$ on a general graph. In our case, with $n$ nodes and $n - 1$ edges (assuming a tree graph), BFS has complexity $\mathcal{O}(n + (n - 1)) = \mathcal{O}(2n - 1) = \mathcal{O}(n)$. The eigendecomposition of a symmetric matrix in the worst case has a complexity of $\mathcal{O}(n^3)$. Therefore, the combined time complexity is $\mathcal{O}(n + n^3) = \mathcal{O}(n^3)$. However, because we only compute the smallest $k_{\max} \ll n$ eigenvectors, the time complexity of the eigendecomposition reduces to $\mathcal{O}(k_{\max} n^2)$. Thus, the overall complexity becomes $\mathcal{O}(n + k_{\max} n^2) = \mathcal{O}(k_{\max} n^2)$, since $k_{\max} n^2$ dominates $n$.

The recursive partitioning process starts with an eigendecomposition on the full graph, which has a time complexity of $\mathcal{O}(k_{\max} n^2)$ for extracting the smallest $k_{\max}$ eigenvectors. This initial computation is the primary contributor to the algorithm's time complexity. As the partitioning proceeds, each recursive call operates on progressively smaller subgraphs. While each subgraph requires an eigendecomposition, the cost decreases significantly as the graph size is reduced at each recursion level. For balanced partitioning, the cumulative cost of these recursive steps remains asymptotically bounded by the initial computation on the full graph. Thus, the overall complexity is effective $\mathcal{O}(k_{\max} n^2)$, ensuring computational feasibility even for large graphs.

## C    Graph Partitioning with Normalised Cut

We recall here the Normalised Cut (NCut) introduced by Shi and Malik [74] for measuring the goodness of a graph partition.

Let $G = (V, E, w)$ be a weighted undirected graph, having $n$ nodes, $V = \{v_i\}_{i=1}^n$, representing points $v_i \in \mathbb{R}^d$. The weight on each edge of the graph $w_{ij} = w(v_i, v_j)$ is a function of the similarity between nodes $v_i$ and $v_j$ and defines an element of the adjacency matrix $W = [w_{ij}] \in [0, 1]^{n \times n}$. The symmetrically normalised Laplacian of the graph [59] is defined as, $L = D^{-1/2}(D - W)D^{-1/2}$, with $D = \operatorname{diag}[d_i] \in \mathbb{R}^{n \times n}$ the diagonal degree matrix and $d_i = \sum_j w_{ij}$.

The normalised cut objective aims to partition the set $V$ into two disjoint sets $A \subset V$ and $B \subset V$ ($A \cup B = V$, $A \cap B = \emptyset$) while minimising the degree of similarity between the two sets and maximising the one within each set, and it is defined as:

$$\operatorname{NCut}(A, B) \coloneqq \frac{\operatorname{cut}(A, B)}{\operatorname{assoc}(A, V)} + \frac{\operatorname{cut}(A, B)}{\operatorname{assoc}(B, V)} \tag{12}$$

where $\operatorname{cut}(A, B) = \sum_{u \in A, v \in B} w(u, v)$ measures the degree of similarity between $A$ and $B$ and is equal to the total weight of edges that the partitioning has removed, $\operatorname{assoc}(A, V) = \sum_{u \in A, t \in V} w(u, t)$ measures the degree of similarity between $A$ and $V$, and $\operatorname{assoc}(B, V)$ is equivalently defined. The NCut is an unbiased measure of the normalised total similarity of the two sets of points. Indeed, normalisation avoids unnatural bias when partitioning out small sets of points. Minimising exactly Equation (12) is NP-complete, according to the proof due to Papadimitriou [74].

However, [74] shows a tractable real-valued solution to the relaxed problem in Equation (12) can be obtained by solving the generalized eigenvalue system $(D - W)x = \lambda Dx$, for $x \in \mathbb{R}^n$ and $\lambda \in \mathbb{R}$. The eigenvectors $x_i$ span an orthogonal basis for functions on $G$ [81]:

$$x_i = \underset{x \in \mathbb{R}^n, \|x\|=1, x \perp x_{<i}}{\arg\min} x^\top L x, \quad \text{with } i = 1, \ldots, n-1 \tag{13}$$

with $x_0 = \mathbb{1} \in \mathbb{R}^n$. The eigenvalues $\lambda_i$, with $0 = \lambda_0 \le \lambda_1 \le \cdots \le \lambda_{n-1}$, are the values of the right-hand side of the above expression. The eigenvector $x_1$ corresponding to the second smallest eigenvalue $\lambda_1$ of $L$ is the non-trivial solution to the quadratic form in Equation (13), called the *Fiedler vector* [29]. However, this real-valued solution can be transformed into a discrete form to partition the set $V$ in two disjoint sets $A$ and $B$ approximating the solution to the normalised cut problem in Equation (12) [74].

The former approach can be expanded to further partition the generated subgraphs, employing the subsequent eigenvectors. Indeed, the eigenvector $x_2$ linked to the third smallest eigenvalue $\lambda_2$ efficiently divides into two parts, $A$ and $B$. However, the practical application reveals that if higher eigenvectors are used, the gap between real-valued and discrete-valued solutions widens, asking for a global mutual orthogonality constraint for all eigenvectors. Consequently, solutions relying on higher eigenvectors tend to be less dependable. It is often more advantageous to restart the partitioning process for each subgraph separately [74].

## D  More about Experiments and Metrics

The code implementation is in `Python 3`. We ran experiments on an ASUS ESC8000 server with two AMD EPYC 7413 24-core processors and 256GB RAM. We used the `PyTorch 2.3` deep learning framework and 2 NVIDIA A6000 GPUs with 48GB of VRAM to accelerate the feature extraction stage. For all the experiments in the paper, we ran our algorithm with the Python multi-threading library `joblib` up to 96 workers.

### D.1  Datasets

**PascalVOC2012 [27]** is a generic object-centric semantic segmentation dataset of 20 object categories and a background class. It consists of $1,449$ images for validation. We follow [12] to obtain a larger image set with additional annotations [34], resulting in $10,582$ images.

**COCO-Stuff [10]** is a complex scene-centric dataset that extends the object-centric MSCOCO [53] dataset, with a high-level hierarchical structure. Concepts are split at the root level into things and stuff, each having outdoor and indoor subsets. There are 12 things and 15 stuff supercategories, and 80 things and 91 stuff categories. Objects appear in complex scenes, with more thing objects per image than PascalVOC2012 (7.3 vs. 2.3). Following [78, 84], we use val2017 split of $5,000$ images.

**Potsdam, Vaihingen [30]** are scene-centric datasets for aerial scene parsing with 6 categories (*roads*, *cars*, *vegetation*, *trees*, *buildings*, *clutter*). The raw $6000 \times 6000$ images are divided into 100 RGB $600 \times 600$ patches. We obtain a total of $3,800$ images for Potsdam and $3,300$ images for Vaihingen.

**Cityscapes [21]** is an urban street scene-centric dataset with a high-level hierarchical structure, having 7 supercategories subdivided into 19 stuff and object categories. Unlike COCO-Stuff and PascalVOC2012, where classes appear in many scene contexts, Cityscapes contains similar street scenes that cover almost all 19 categories. The test split has $500$ images.

**Mapillary Vistats [60]** is an urban street scene-centric dataset with a high-level hierarchical structure, having 6 root-level categories subdivided into 37 supercategories and finally 66 categories. Unlike Cityscapes and KITTI-STEP, street scenes are captured from various environments worldwide, including several countries, weather conditions, and seasons. It aims to provide a diverse set of street scenes. The validation split has $2,000$ images.

**KITTI-STEP [89], KITTI-SS [1]** are datasets for urban scene understanding, instance detection, and object tracking. They have the same categories as Cityscapes and the same hierarchical structure. There are $2,981$ validation frames of KITTI-STEP, 200 test images of KITTI-SS.

**Pascal-Part [15]** is an extension of the PascalVOC2010 [27] dataset, designed specifically for fine-grained annotations of objects. It is a part-centric dataset with a low-level hierarchical structure. It contains 20 object classes, each subdivided into low-level parts (i.e., *head*, *left/right-eye*, *torso*, *left/right-arm*, etc., for category *person*) for a total of 198 distinct part classes and a background category. The dataset contains $10,103$ images.

**PartImageNet [35]** is a part-centric dataset with a low-level hierarchical structure designed for fine-grained part segmentation. It extends the ImageNet dataset by providing detailed part annotations for a subset of the images. It has 11 object supercategories and 39 part supercategories. **PartImagenet-158 [35]** arranges the annotation by ImageNet categories, counting 158 object classes and a total of 597 part categories. The validation set contains a total of $2,957$ images.

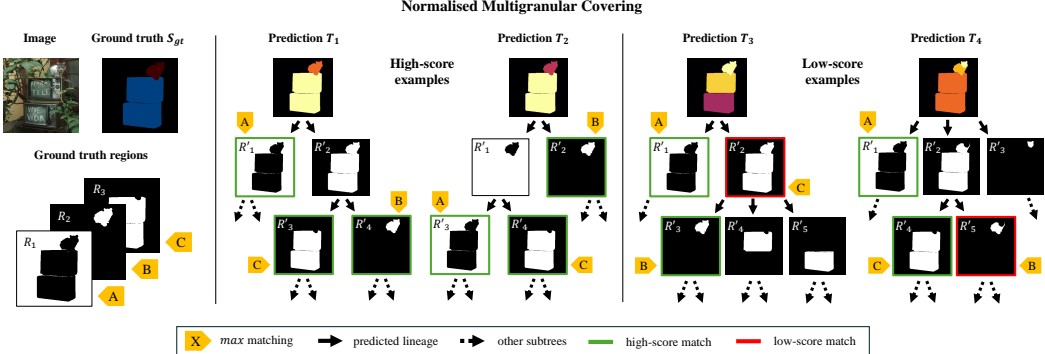

Figure 7: **Normalised Multigranular Covering** (NMCovering**) examples.** For each available ground truth categorical region $R$ in the semantic map $S_{gt}$ (left), we evaluate the overlap with the unrolled segments $R'$ in the predicted region tree, e.g. $T_1$. The yellow labels indicate the maximum IoU matching correspondence between the ground truth and the prediction. Green line borders indicate high-score matching and red line borders indicate low-score matching. We propose two high-scoring predictions (centre) and two low-scoring (right). The total NMCovering is the average sum of the matching scores, as defined in Equation (3). The NMCovering metric evaluates the granularity-independent performance of the semantic segmentation model. The absence of correct semantic regions in $T_3$ and $T_4$ yields low score matches; see plate $C$ in $T_3$ and plate $B$ in $T_4$.

## D.2 Discussion About The Metrics

We average the metrics over each image in the dataset, ensuring a comprehensive assessment across varying image contexts. In the hierarchical scenario, the score gives equal importance to all levels, recognising the significance of coarse and fine-grained segmentation. This approach reflects the nuanced structure of hierarchical data, where higher and lower granularity levels provide complementary insights.

The *maximum overlap heuristics* we use do not enforce exclusive matching. We intentionally choose this method to accommodate scenarios where regions in subsequent hierarchical levels overlap. For instance, in an image of human hands, the coarse category *person* may overlap with the finer category *hand*. An exclusive matching strategy might misinterpret this overlap, leading to an inaccurate assessment of segmentation performance. Our approach acknowledges such overlaps, providing a more realistic evaluation.

Additionally, the granularity of ground-truth annotations is often limited. To address this, our evaluation process disregards predictions that do not best match any annotated object, treating them as neither true positives nor false positives. This avoids penalising the model for predicting more detailed segments than the available annotations. Moreover, this approach permits the evaluation of the model according to standard category micro and macro-averaged segmentation metrics [56] such as the micro *pixel Accuracy* (pAcc) and the macro *mean pixel Accuracy* (mAcc), the per-class *Intersection over Union* (IoU) and the relative macro *mean IoU* (mIoU), and micro *frequency weighted IoU* (fIoU).

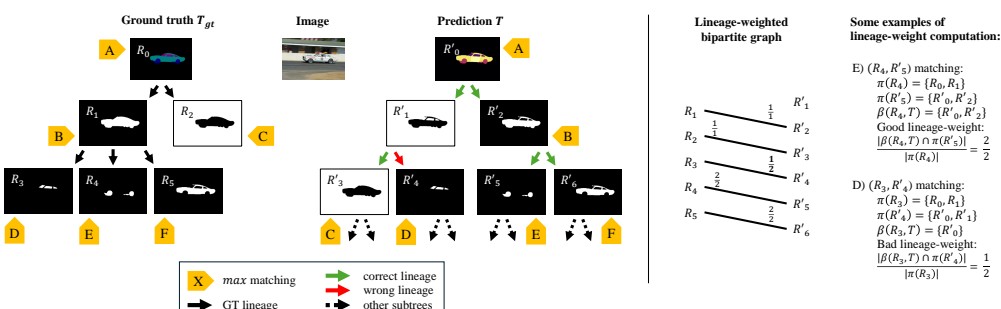

**Normalised Hierarchical Covering**

Figure 8: **Normalised Hierarchical Covering (**NHCovering**) computation example.** Given the semantic tree $T_{gt}$ (left), for each available ground truth categorical region $R$, we evaluate the overlap with the unrolled segments $R'$ in the predicted region tree $T$. We consider one low-score lineage prediction edge $(R'_1, R'_4)$ and one high-score $(R'_2, R'_5)$. The yellow labels indicate the maximum IoU matching correspondence between the ground truth and the predicted regions. Green and red arrows indicate correct and wrong lineage prediction, respectively. The total NHCovering is the sum of the matching scores weighted by the ratio of correct lineages, as reported in Equation (4). The NHCovering metric assesses the granularity and hierarchy-independent performance of the semantic segmentation model. Examples of lineage-weight computation are reported for the $E$ and $D$ matching, on the right, using the operators $\pi(\cdot)$ and $\beta(\cdot, \cdot)$ defined in Section 4.1.

Interestingly, the NMCovering metric can define an upper bound for the NHCovering score with appropriate modifications. Specifically, we find that $\text{NHCovering}(T \to T_{gt}) \leq \text{NMCovering}(T \to T_{gt})$. Equality between these metrics would indicate optimal model performance in terms of hierarchical inclusion, suggesting that the model not only segments accurately but also respects the hierarchical structure of the ground truth. Figures 7 and 8 offers finer insight into the metrics purpose.

These metrics provide a robust framework for evaluating hierarchical segmentation models, balancing granularity and hierarchical accuracy while accounting for the inherent complexities in real-world image data.

## D.3 Quantitative Results

Our hierarchical semantic segmentation algorithm demonstrates strong performance across multiple datasets and hierarchical segmentation tasks, effectively capturing meaningful semantic structures at various levels of granularity.

Table 7 showcases the comparative strength of our method, achieving high NMCovering scores against other low-level pixel hierarchical clustering algorithms. With $\lambda_{\max} = 0.6$, our approach effectively segments the validation set into hierarchical clusters that closely align with ground-truth structures, indicating robustness in preserving semantic hierarchies.

Table 8 further illustrate our algorithm's adaptability in aerial scene segmentation, achieving notable NMCovering scores on both train sets. Using DINO-ViT-B8 features with $\lambda_{\max} = 0.9$, the algorithm accurately segments six primary categories, confirming its applicability to complex geospatial datasets where hierarchical segmentation is critical.

Tables 9 and 10 highlight our model's proficiency in urban scene segmentation, achieving high NHCovering and mIoU scores across 19 categories and seven supercategories. Notably, the use of hierarchical labels in Table 9 and detailed category IoUs in Table 10 show the model's capability to distinguish fine-grained features within broader urban contexts. This performance suggests our approach can effectively capture semantic structures across varying levels of granularity, making it well-suited for dynamic urban environments.

Table 13 is a challenging benchmark with its extensive hierarchy of 'thing' and 'stuff' categories, totalling 182 classes and 27 supercategories. Our algorithm achieves competitive performance across

both high-level supercategories and fine-grained categories, achieving strong IoU scores across each subset. This indicates that the algorithm effectively balances both coarse and fine semantic segmentation layers, enabling nuanced representation across diverse objects and materials.

In Table 11, focused on Conditional Random Field (CRF) post-processing, our method shows that including CRF enhances segmentation accuracy across various datasets, validating its role in refining boundaries for complex regions. Additionally, comparisons with non-hierarchical spectral clustering methods Table 12 underscore our approach's advantage in multi-level segmentation, achieving higher NMCovering scores and consistent performance even without the need for CRF post-processing.

Table 7: **Boundary potential vs. semantic smoothness.** Comparison among hierarchical clustering algorithms in terms of NMCovering on PascalVOC2012 *val* set for $\lambda_{max} = 0.6$.

| Method | bkgd | airplane | bicycle | bird | boat | bottle | bus | car | cat | chair | cow | d. table | dog | horse | bike | person | p. plant | sheep | couch | train | tv | mIoU | pAcc | mAcc | fIoU | NMCovering |
|---|---|---|---|---|---|---|---|---|---|---|---|---|---|---|---|---|---|---|---|---|---|---|---|---|---|---|
| SE-OWT-UCM [24] | 71.3 | 56.4 | 17.9 | 46.4 | 49.4 | 36.4 | 55.1 | 42.6 | 67.8 | 33.3 | 54.0 | 53.2 | 56.2 | 46.6 | 41.5 | 43.7 | 28.4 | 50.2 | 59.0 | 55.0 | 51.3 | 48.4 | 83.0 | 79.4 | 65.5 | 59.0 |
| PMI-OWT-UCM [40] | 75.4 | 55.4 | 14.0 | 45.2 | 46.4 | 33.4 | 48.9 | 36.8 | 71.9 | 35.1 | 57.6 | 50.6 | 60.2 | 43.0 | 37.4 | 39.4 | 21.4 | 58.9 | 60.3 | 50.5 | 45.4 | 47.0 | **86.5** | 81.6 | 68.1 | 61.3 |
| **Ours** | 76.9 | 73.4 | 51.0 | 82.1 | 72.4 | 82.5 | 85.6 | 81.1 | 90.2 | 71.2 | 87.1 | 68.8 | 87.7 | 78.3 | 79.2 | 82.1 | 70.8 | 84.7 | 82.9 | 82.9 | 68.8 | **78.1** | 82.6 | **91.2** | **78.1** | **75.4** |

Table 8: **Hierarchical semantic segmentation on Potsdam and Vaihingen *train* sets.** We use DINO-ViT-B8 [11] features and $\lambda_{max} = 0.9$. The two datasets have six categories. Segmentation performances are computed using NMCovering for ground truth masks exclusive matching.

| Dataset | roads | buildings | vegetation | tree | car | clutter | mIoU | pAcc | mAcc | fIoU | NHCovering |
|---|---|---|---|---|---|---|---|---|---|---|---|
| Potsdam | 68.3 | 76.4 | 61.5 | 53.4 | 46.9 | 47.3 | 59.0 | 83.4 | 83.2 | 65.0 | 57.9 |
| Vaihingen | 56.7 | 67.7 | 43.3 | 63.4 | 29.6 | 56.4 | 52.8 | 76.8 | 73.9 | 58.1 | 53.9 |

Table 9: **Hierarchical semantic segmentation on Cityscapes and KITTI-STEP *val* sets and KITTI-SS *train* set.** We use DINO-ViT-B8 [11] features and $\lambda_{max} = 0.8$. The three datasets have 19 valid categories from the Cityscapes dataset divided into seven supercategories. Segmentation performances are computed using NHCovering for ground truth masks exclusive matching.

| Dataset | flat | construction | object | nature | sky | human | vehicle | mIoU$^S$ | pAcc$^S$ | mAcc$^S$ | fIoU$^S$ | NMCovering$^S$ |
|---|---|---|---|---|---|---|---|---|---|---|---|---|
| Cityscapes | 83.3 | 63.6 | 35.4 | 61.5 | 61.0 | 43.7 | 67.1 | 59.4 | 82.3 | 74.8 | 70.9 | 55.3 |
| KITTI-STEP | 74.3 | 58.7 | 38.1 | 65.9 | 72.7 | 44.4 | 73.0 | 61.0 | 79.3 | 76.7 | 67.8 | 56.1 |
| KITTI-SS | 73.4 | 59.7 | 38.5 | 64.5 | 73.4 | 26.5 | 69.8 | 58.0 | 78.5 | 75.8 | 67.2 | 57.4 |

| Dataset | road | sidewalk | building | wall | fence | pole | traffic light | traffic sign | vegetation | terrain | sky | person | rider | car | truck | bus | train | motorcycle | bicycle | mIoU | pAcc | mAcc | fIoU | NMCovering |
|---|---|---|---|---|---|---|---|---|---|---|---|---|---|---|---|---|---|---|---|---|---|---|---|---|
| Cityscapes | 83.9 | 48.8 | 64.7 | 47.2 | 46.9 | 27.7 | 15.3 | 29.7 | 61.7 | 41.4 | 61.0 | 43.7 | 20.5 | 70.6 | 54.3 | 63.2 | 25.1 | 41.6 | 48.0 | 82.8 | 76.2 | 68.8 | 44.8 |
| KITTI-STEP | 75.0 | 50.5 | 61.2 | 56.2 | 50.0 | 27.4 | 26.1 | 33.4 | 67.1 | 56.5 | 72.7 | 43.8 | 20.4 | 75.3 | 52.7 | 56.9 | 80.7 | 25.1 | 41.0 | 51.1 | 79.9 | 76.5 | 65.7 | 48.4 |
| KITTI-SS | 73.2 | 46.3 | 61.1 | 60.8 | 50.5 | 29.1 | 26.6 | 37.1 | 64.4 | 58.9 | 73.4 | 27.5 | 18.5 | 72.0 | 37.6 | 27.8 | 63.1 | 28.7 | 20.2 | 46.2 | 78.6 | 75.0 | 65.4 | 49.3 |

Table 10: **Hierarchical semantic segmentation on Cityscapes and KITTI-STEP *val* sets and KITTI-SS *train* set.** We use DINO-ViT-B8 [11] features and $\lambda_{max} = 0.8$. The three datasets have 19 valid categories from the Cityscapes dataset. Segmentation performances are computed using NHCovering for exclusive matching of predictions with ground truth masks. From the second column, we show the relative category IoU. Last four columns show mean IoU (mIoU), frequency weighted IoU (fIoU), pixel accuracy (pAcc) and mean accuracy (mAcc).

| Dataset | road | sidewalk | building | wall | fence | pole | traffic light | traffic sign | vegetation | terrain | sky | person | rider | car | truck | bus | train | motorcycle | bicycle | mIoU | pAcc | mAcc | fIoU | NHCovering |
|---|---|---|---|---|---|---|---|---|---|---|---|---|---|---|---|---|---|---|---|---|---|---|---|---|
| Cityscapes | 81.5 | 48.8 | 58.6 | 46.1 | 45.7 | 25.9 | 12.3 | 26.3 | 55.8 | 39.1 | 50.9 | 33.7 | 11.4 | 63.7 | 51.8 | 61.8 | 63.2 | 19.4 | 32.5 | 46.6 | 77.8 | 72.4 | 65.5 | 34.8 |
| KITTI-STEP | 68.5 | 49.3 | 53.8 | 54.4 | 47.1 | 25.9 | 24.8 | 29.6 | 62.8 | 56.4 | 58.7 | 36.0 | 14.7 | 65.1 | 44.5 | 50.5 | 79.7 | 18.1 | 34.7 | 48.5 | 73.0 | 72.3 | 60.9 | 38.0 |
| KITTI-SS | 68.6 | 46.3 | 54.2 | 55.9 | 49.3 | 28.4 | 25.0 | 29.6 | 60.6 | 59.1 | 59.0 | 16.9 | 15.6 | 63.1 | 35.5 | 24.9 | 63.1 | 28.7 | 10.7 | 44.9 | 73.4 | 71.5 | 61.2 | 39.1 |

Table 11: **CRF ablation.** We use maximum overlap for ground-truth category matching.

| Dataset (mIoU) | w/o CRF | w CRF |
|---|---|---|
| Cityscapes | 48.8 | 51.0 |
| KITTI-STEP | 51.2 | 53.4 |
| Mapillary Vistas | 47.6 | 48.5 |
| Potsdam | 58.9 | 63.2 |

Table 12: **Recursive vs. simultaneous on PascalVOC2012 *val* set.** Comparison between deep recursive (Ours) and simultaneous spectral clustering (Melas-Kyriazi et al. [58]) for $m = \{4, 8, 16\}$ using a maximum overlap for category matching in each image. All the experiments run on pre-extracted features with DINO-ViT-S8 [11], without CRF. The other parameters are defaulted in Section 4. Notice that for simultaneous clustering the NMCovering equals the NFCovering [44].

| Method | bkgd | airplane | bicycle | bird | boat | bottle | bus | car | cat | chair | cow | d. table | dog | horse | bike | person | p. plant | sheep | couch | train | tv | mIoU | pAcc | mAcc | fIoU | NMCovering |
|---|---|---|---|---|---|---|---|---|---|---|---|---|---|---|---|---|---|---|---|---|---|---|---|---|---|---|
| [58] ($m = 4$) | 39.4 | 47.7 | 23.6 | 35.6 | 36.1 | 26.4 | 46.5 | 31.6 | 45.6 | 25.3 | 45.1 | 43.2 | 41.8 | 37.2 | 45.7 | 35.5 | 21.0 | 42.5 | 45.5 | 47.0 | 20.6 | 36.4 | 45.5 | 60.0 | 39.3 | 40.7 |
| Ours ($m = 4$) | 54.8 | 34.4 | 13.0 | 23.5 | 25.6 | 20.7 | 50.6 | 25.0 | 48.8 | 18.8 | 40.3 | 29.0 | 36.5 | 40.1 | 39.1 | 31.8 | 15.2 | 34.9 | 36.8 | 41.1 | 18.3 | 32.3 | 62.5 | 68.4 | 49.4 | 44.8 |
| [58] ($m = 8$) | 27.8 | 44.2 | 32.1 | 42.3 | 39.6 | 26.2 | 29.9 | 31.9 | 34.5 | 37.0 | 35.3 | 37.0 | 35.5 | 37.7 | 39.8 | 36.5 | 30.7 | 34.2 | 40.2 | 35.1 | 39.9 | 35.5 | 31.5 | 42.6 | 29.9 | 36.4 |
| Ours ($m = 8$) | 61.1 | 51.0 | 22.1 | 42.0 | 42.6 | 34.8 | 69.5 | 46.6 | 70.0 | 29.3 | 64.1 | 41.3 | 58.4 | 53.1 | 56.1 | 43.6 | 24.1 | 57.7 | 54.5 | 63.3 | 34.8 | 48.6 | 69.1 | 76.4 | 58.5 | 55.5 |
| [58] ($m = 16$) | 16.0 | 28.7 | 33.0 | 29.3 | 30.9 | 23.6 | 17.9 | 24.0 | 21.4 | 33.5 | 22.2 | 28.1 | 22.0 | 24.3 | 25.1 | 27.5 | 31.0 | 23.5 | 29.1 | 21.2 | 34.2 | 26.0 | 19.0 | 27.8 | 18.5 | 27.7 |
| Ours ($m = 16$) | 65.2 | 66.8 | 33.7 | 58.7 | 51.9 | 49.8 | 76.1 | 58.8 | 78.1 | 39.4 | 70.0 | 50.8 | 72.2 | 67.3 | 65.2 | 56.9 | 32.9 | 64.3 | 60.2 | 71.6 | 42.5 | 58.6 | 72.1 | 78.9 | 64.3 | 62.2 |
| Ours ($m = 100$) | 69.7 | 83.1 | 51.7 | 85.8 | 75.2 | 70.2 | 84.0 | 82.0 | 86.7 | 67.1 | 85.8 | 66.3 | 85.8 | 80.0 | 76.5 | 73.5 | 66.3 | 86.4 | 81.3 | 75.9 | 66.9 | 76.2 | 76.8 | 85.6 | 72.0 | 72.5 |

Table 13: **Hierarchical semantic segmentation on the COCO-Stuff *val* set.** We use DINO-ViT-B8 [11] features and $\lambda_{max} = 0.6$. COCO-Stuff dataset has 91 'thing' categories (inherited from MSCOCO), 91 'stuff' categories and 27 supercategories. Segmentation performances are computed using NHCovering for exclusive matching of predictions with ground truth masks. The rows show the hierarchical structure of COCO-Stuff. The first column shows the separation between things/stuff. The second column shows supercategory labels (Coarse Tags), the third column shows the relative supercategory IoU ($\text{IoU}^S$) when considering all supercategories together, and the fourth column considers 12 things supercategories only, and the fifth column considers 15 stuff supercategories only. Sixth column shows leaf labels (Fine Tags), seventh column shows relative category IoU ($\text{IoU}$) when considering all categories, the eighth column considers 91 thing categories only and the ninth column considers 91 stuff categories only. Last row shows mean IoU ($\text{mIoU}$), frequency weighted IoU ($\text{fIoU}$), pixel accuracy ($\text{pAcc}$), and mean accuracy ($\text{mAcc}$) for each experiment. Here, accuracy values are reported in the decimal range $[0, 1]$.

Left half:

| Set | Coarse Tags | Things & Stuff | Things-only | Stuff-only | Fine Tags | Things & Stuff | Things-only | Stuff-only |
|---|---|---|---|---|---|---|---|---|
| thing | persons | 0.66 | 0.66 | | person | 0.65 | 0.66 | |
| | vehicles | 0.7 | 0.7 | | bicycle | 0.41 | 0.5 | |
| | | | | | car | 0.44 | 0.52 | |
| | | | | | motorcycle | 0.67 | 0.69 | |
| | | | | | airplane | 0.69 | 0.69 | |
| | | | | | bus | 0.76 | 0.77 | |
| | | | | | train | 0.75 | 0.75 | |
| | | | | | truck | 0.61 | 0.64 | |
| | | | | | boat | 0.56 | 0.55 | |
| | outdoors | 0.56 | 0.6 | | traffic light | 0.2 | 0.28 | |
| | | | | | fire hydrant | 0.64 | 0.62 | |
| | | | | | street sign | N/A | N/A | |
| | | | | | stop sign | 0.64 | 0.67 | |
| | | | | | parking meter | 0.61 | 0.72 | |
| | | | | | bench | 0.55 | 0.6 | |
| | animals | 0.78 | 0.78 | | bird | 0.59 | 0.61 | |
| | | | | | cat | 0.81 | 0.81 | |
| | | | | | dog | 0.73 | 0.76 | |
| | | | | | horse | 0.67 | 0.69 | |
| | | | | | sheep | 0.78 | 0.78 | |
| | | | | | cow | 0.78 | 0.78 | |
| | | | | | elephant | 0.79 | 0.79 | |
| | | | | | bear | 0.84 | 0.84 | |
| | | | | | zebra | 0.82 | 0.82 | |
| | | | | | giraffe | 0.75 | 0.75 | |
| | accessorys | 0.48 | 0.51 | | hat | N/A | N/A | |
| | | | | | backpack | 0.13 | 0.17 | |
| | | | | | umbrella | 0.69 | 0.7 | |
| | | | | | shoe | N/A | N/A | |
| | | | | | eyeglasses | N/A | N/A | |
| | | | | | handbag | 0.11 | 0.15 | |
| | | | | | tie | 0.15 | 0.16 | |
| | | | | | suitcase | 0.63 | 0.65 | |
| | sportss | 0.37 | 0.37 | | frisbee | 0.26 | 0.38 | |
| | | | | | skis | 0.22 | 0.24 | |
| | | | | | snowboard | 0.33 | 0.33 | |
| | | | | | sports ball | 0.09 | 0.07 | |
| | | | | | kite | 0.4 | 0.49 | |
| | | | | | baseball bat | 0.1 | 0.13 | |
| | | | | | baseball glove | 0.19 | 0.19 | |
| | | | | | skateboard | 0.43 | 0.44 | |
| | | | | | surfboard | 0.48 | 0.46 | |
| | | | | | tennis racket | 0.29 | 0.27 | |
| | kitchens | 0.54 | 0.56 | 0.72 | bottle | 0.23 | 0.35 | |
| | | | | | plate | N/A | N/A | |
| | | | | | wine glass | 0.28 | 0.38 | 0.72 |
| | | | | | cup | 0.32 | 0.38 | |
| | | | | | fork | 0.12 | 0.23 | |
| | | | | | knife | 0.1 | 0.16 | |
| | | | | | spoon | 0.13 | 0.15 | |
| | | | | | bowl | 0.58 | 0.62 | |
| | foods | 0.74 | 0.74 | | banana | 0.68 | 0.73 | |
| | | | | | apple | 0.42 | 0.45 | |
| | | | | | sandwich | 0.69 | 0.69 | |
| | | | | | orange | 0.64 | 0.68 | |
| | | | | | broccoli | 0.61 | 0.61 | |
| | | | | | carrot | 0.57 | 0.6 | |
| | | | | | hot dog | 0.69 | 0.71 | |
| | | | | | pizza | 0.81 | 0.83 | |
| | | | | | donut | 0.78 | 0.78 | |
| | | | | | cake | 0.72 | 0.69 | |
| | furnitures | 0.69 | 0.69 | | chair | 0.44 | 0.46 | |
| | | | | | couch | 0.66 | 0.68 | |
| | | | | | potted plant | 0.43 | 0.51 | |
| | | | | | bed | 0.77 | 0.77 | |
| | | | | | mirror | N/A | N/A | |
| | | | | | dining table | 0.72 | 0.73 | |
| | | | | | window | N/A | N/A | |
| | | | | | desk | N/A | N/A | |
| | | | | | toilet | 0.68 | 0.69 | |
| | | | | | door | N/A | N/A | |
| | electronics | 0.63 | 0.64 | | tv | 0.64 | 0.65 | |
| | | | | | laptop | 0.64 | 0.65 | |
| | | | | | mouse | 0.11 | 0.19 | |
| | | | | | remote | 0.2 | 0.27 | |
| | | | | | keyboard | 0.63 | 0.62 | |
| | | | | | cell phone | 0.49 | 0.53 | |
| | appliances | 0.66 | 0.66 | | microwave | 0.51 | 0.61 | |
| | | | | | oven | 0.62 | 0.62 | |
| | | | | | toaster | 0.48 | 0.56 | |
| | | | | | sink | 0.34 | 0.48 | |
| | | | | | refrigerator | 0.7 | 0.7 | |
| | indoors | 0.59 | 0.61 | | blender | N/A | N/A | |
| | | | | | book | 0.39 | 0.47 | |
| | | | | | clock | 0.44 | 0.5 | |
| | | | | | vase | 0.39 | 0.5 | |
| | | | | | scissors | 0.48 | 0.57 | |
| | | | | | teddy bear | 0.75 | 0.76 | |
| | | | | | hair drier | 0.46 | 0.48 | |
| | | | | | toothbrush | 0.14 | 0.21 | |
| | | | | | hair brush | N/A | N/A | |
| | $\text{mIoU}^S$ | 0.64 | 0.64 | 0.66 | mIoU | 0.55 | 0.55 | 0.59 |
| | $\text{fIoU}^S$ | 0.69 | 0.78 | 0.71 | fIoU | 0.66 | 0.78 | 0.69 |

Right half:

| Set | Coarse Tags | Things & Stuff | Things-only | Stuff-only | Fine Tags | Things & Stuff | Things-only | Stuff-only |
|---|---|---|---|---|---|---|---|---|
| stuff | textile | 0.45 | | 0.47 | banner | 0.47 | | 0.48 |
| | | | | | blanket | 0.52 | | 0.53 |
| | | | | | cloth | 0.32 | | 0.43 |
| | | | | | clothes | 0.18 | | 0.21 |
| | | | | | curtain | 0.64 | | 0.65 |
| | | | | | mat | 0.45 | | 0.49 |
| | | | | | napkin | 0.42 | | 0.5 |
| | | | | | pillow | 0.25 | | 0.22 |
| | | | | | rug | 0.6 | | 0.61 |
| | | | | | textile-other | 0.38 | | 0.43 |
| | | | | | towel | 0.41 | | 0.44 |
| | plant | 0.73 | | 0.73 | branch | 0.54 | | 0.54 |
| | | | | | bush | 0.55 | | 0.55 |
| | | | | | flower | 0.42 | | 0.47 |
| | | | | | grass | 0.75 | | 0.76 |
| | | | | | leaves | 0.64 | | 0.66 |
| | | | | | moss | 0.64 | | 0.65 |
| | | | | | plant-other | 0.51 | | 0.51 |
| | | | | | straw | 0.74 | | 0.74 |
| | | | | | tree | 0.72 | | 0.72 |
| | building | 0.66 | | 0.66 | bridge | 0.5 | | 0.5 |
| | | | | | building-other | 0.64 | | 0.64 |
| | | | | | house | 0.6 | | 0.6 |
| | | | | | roof | 0.49 | | 0.53 |
| | | | | | skyscraper | 0.67 | | 0.67 |
| | | | | | tent | 0.54 | | 0.55 |
| | furniture | 0.54 | | 0.54 | cabinet | 0.57 | | 0.58 |
| | | | | | counter | 0.48 | | 0.53 |
| | | | | | cupboard | 0.58 | | 0.58 |
| | | | | | desk-stuff | 0.58 | | 0.59 |
| | | | | | door-stuff | 0.48 | | 0.5 |
| | | | | | furniture-other | 0.45 | | 0.47 |
| | | | | | light | 0.18 | | 0.18 |
| | | | | | mirror-stuff | 0.48 | | 0.49 |
| | | | | | shelf | 0.48 | | 0.48 |
| | | | | | stairs | 0.41 | | 0.41 |
| | | | | | table | 0.53 | | 0.56 |
| | structural | 0.59 | | 0.59 | cage | 0.6 | | 0.6 |
| | | | | | fence | 0.56 | | 0.57 |
| | | | | | net | 0.59 | | 0.61 |
| | | | | | railing | 0.4 | | 0.41 |
| | | | | | structural-other | 0.47 | | 0.49 |
| | rawmaterial | 0.48 | 0.83 | 0.48 | cardboard | 0.48 | | 0.51 |
| | | | | | metal | 0.39 | | 0.41 |
| | | | | | paper | 0.37 | | 0.4 |
| | | | | | plastic | 0.37 | 0.83 | 0.41 |
| | floor | 0.69 | | 0.7 | carpet | 0.73 | | 0.73 |
| | | | | | floor-marble | 0.69 | | 0.69 |
| | | | | | floor-other | 0.55 | | 0.57 |
| | | | | | floor-stone | 0.74 | | 0.75 |
| | | | | | floor-tile | 0.71 | | 0.72 |
| | | | | | floor-wood | 0.66 | | 0.67 |
| | ceiling | 0.71 | | 0.71 | ceiling-other | 0.69 | | 0.68 |
| | | | | | ceiling-tile | 0.74 | | 0.74 |
| | sky | 0.87 | | 0.87 | clouds | 0.87 | | 0.87 |
| | | | | | sky-other | 0.85 | | 0.85 |
| | water | 0.83 | | 0.83 | fog | 0.86 | | 0.86 |
| | | | | | river | 0.78 | | 0.78 |
| | | | | | sea | 0.84 | | 0.84 |
| | | | | | water-other | 0.77 | | 0.77 |
| | | | | | waterdrops | 0.29 | | 0.29 |
| | food | 0.57 | | 0.58 | food-other | 0.53 | | 0.55 |
| | | | | | fruit | 0.43 | | 0.42 |
| | | | | | salad | 0.58 | | 0.58 |
| | | | | | vegetable | 0.56 | | 0.58 |
| | ground | 0.79 | | 0.79 | dirt | 0.67 | | 0.68 |
| | | | | | gravel | 0.63 | | 0.62 |
| | | | | | ground-other | 0.7 | | 0.72 |
| | | | | | mud | 0.68 | | 0.68 |
| | | | | | pavement | 0.69 | | 0.69 |
| | | | | | platform | 0.58 | | 0.58 |
| | | | | | playingfield | 0.83 | | 0.83 |
| | | | | | railroad | 0.57 | | 0.57 |
| | | | | | road | 0.7 | | 0.71 |
| | | | | | sand | 0.79 | | 0.79 |
| | | | | | snow | 0.85 | | 0.85 |
| | solid | 0.67 | | 0.67 | hill | 0.66 | | 0.66 |
| | | | | | mountain | 0.68 | | 0.68 |
| | | | | | rock | 0.72 | | 0.72 |
| | | | | | solid-other | 0.47 | | 0.5 |
| | | | | | stone | 0.63 | | 0.63 |
| | | | | | wood | 0.51 | | 0.53 |
| | wall | 0.67 | | 0.67 | wall-brick | 0.56 | | 0.57 |
| | | | | | wall-concrete | 0.67 | | 0.68 |
| | | | | | wall-other | 0.59 | | 0.59 |
| | | | | | wall-panel | 0.68 | | 0.68 |
| | | | | | wall-stone | 0.69 | | 0.64 |
| | | | | | wall-tile | 0.63 | | 0.64 |
| | | | | | wall-wood | 0.6 | | 0.61 |
| | window | 0.6 | | 0.6 | window-blind | 0.69 | | 0.69 |
| | | | | | window-other | 0.56 | | 0.57 |
| | $\text{pAcc}^S$ | 0.86 | 0.93 | 0.87 | pAcc | 0.85 | 0.93 | 0.86 |
| | $\text{mAcc}^S$ | 0.85 | 0.87 | 0.84 | mAcc | 0.83 | 0.85 | 0.82 |

## D.4    Qualitative Results

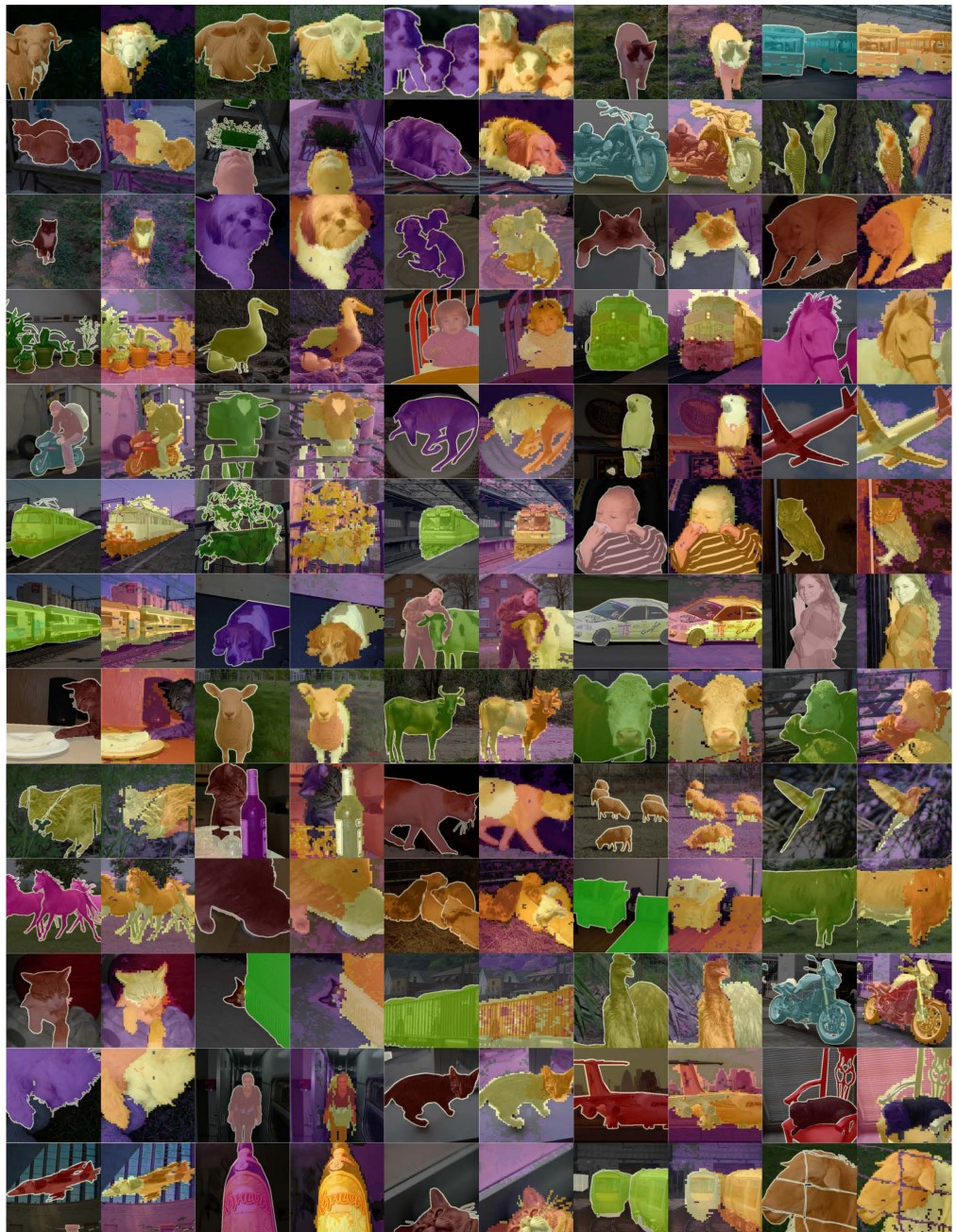

Figure 9: **Qualitative results on PascalVOC2012.** Random sampling from a subset of our results, refined with CRF, having NMCovering greater than 70%. We show predicted subtrees (right) overlapping with semantic masks (left). Heatmap colours encode leaves' distance in the subtrees.

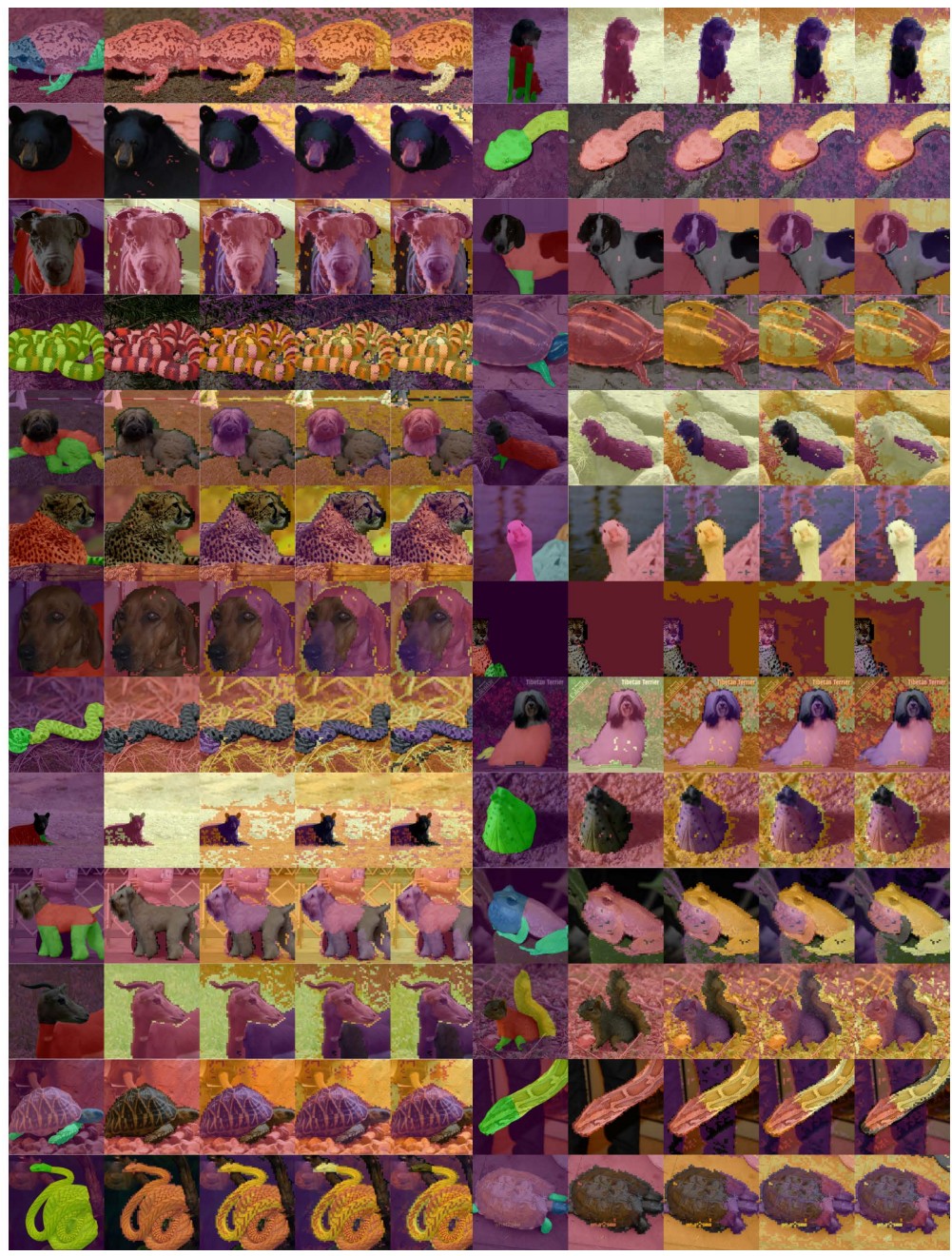

Figure 10: **Qualitative results on PartImageNet.** Random sampling from a subset of our results, refined with CRF, having NHCovering greater than 70%. The left column shows the ground truth part masks. The second to fifth column shows the predicted regions for each tree depth.

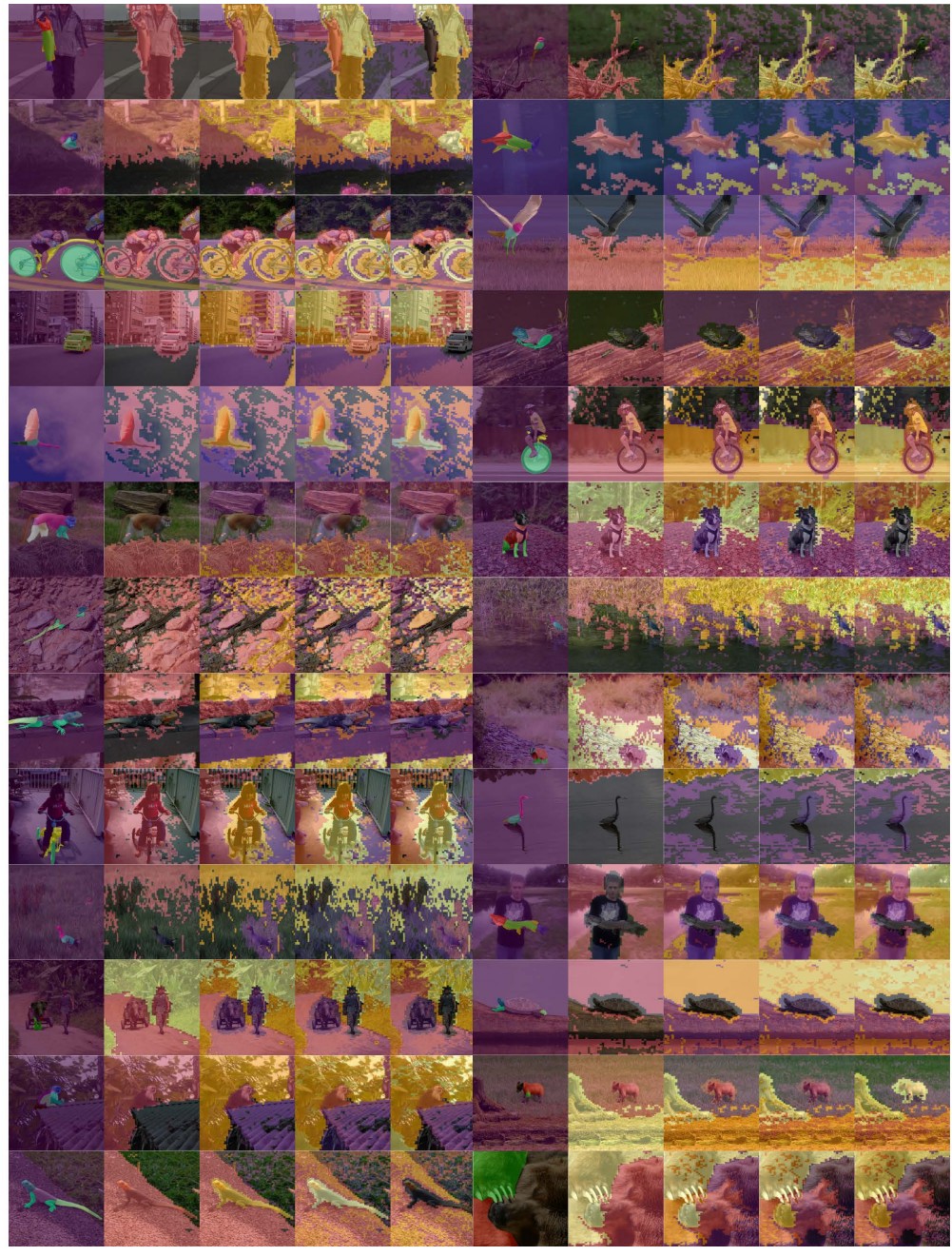

Figure 11: **Qualitative results on PartImageNet.** Random sampling from a subset of our results, refined with CRF, having NHCovering lower than 20%. The image shows failures in identifying very small parts. The left column shows the ground truth part masks. The second to fifth column shows the predicted regions for each tree depth. The prediction colour code does not reflect the ground truth one.

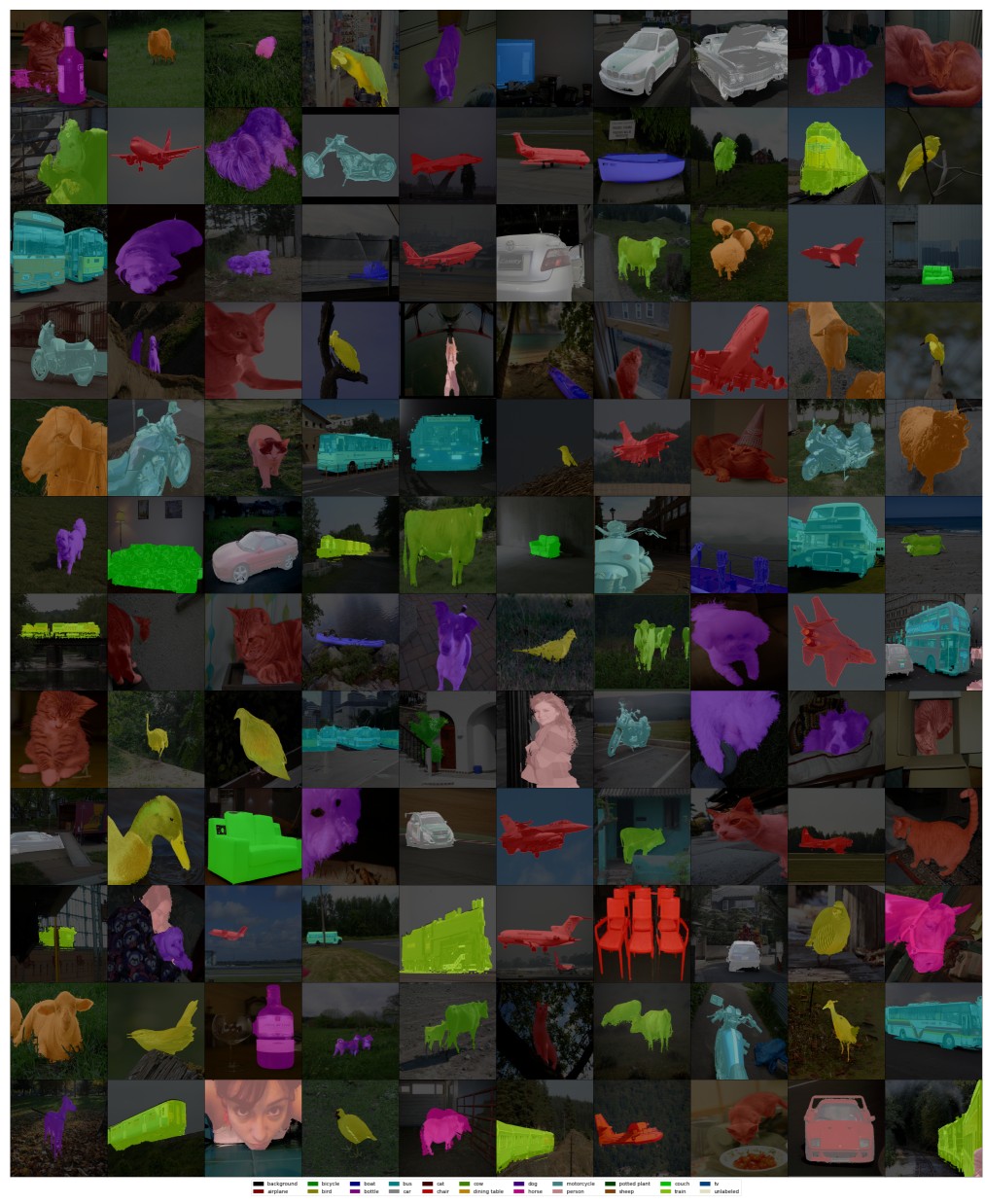

Figure 12: **Qualitative results on PascalVOC2012.** Random sampling from a subset of our results, refined with CRF, having NMCovering greater than 70%. We assign unsupervised masks to the best overlapping classes.

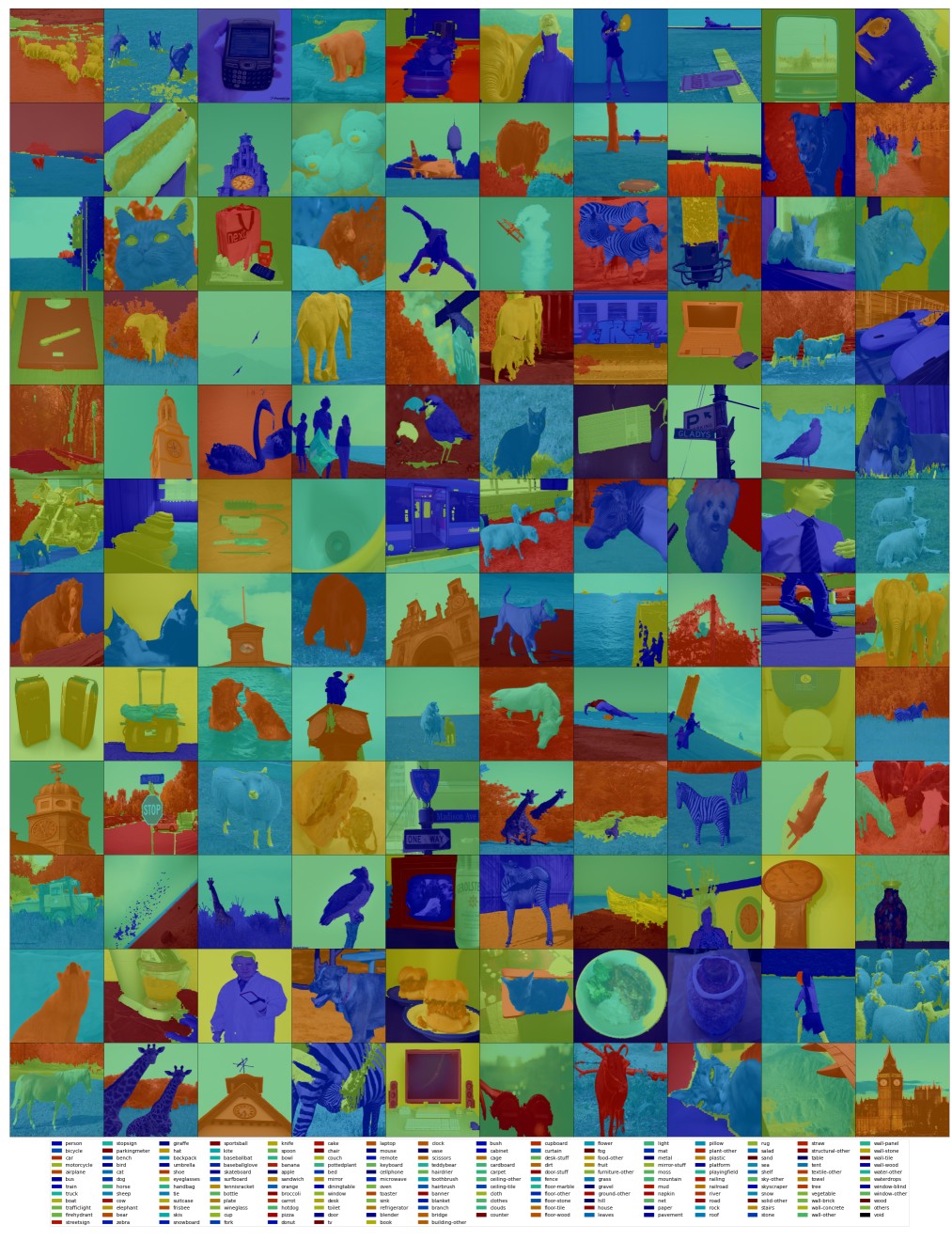

Figure 13: **Qualitative results on COCO-Stuff.** Random sampling from a subset of our results, refined with CRF, with NMCovering greater than 60%. We assign unsupervised masks to the best overlapping classes.

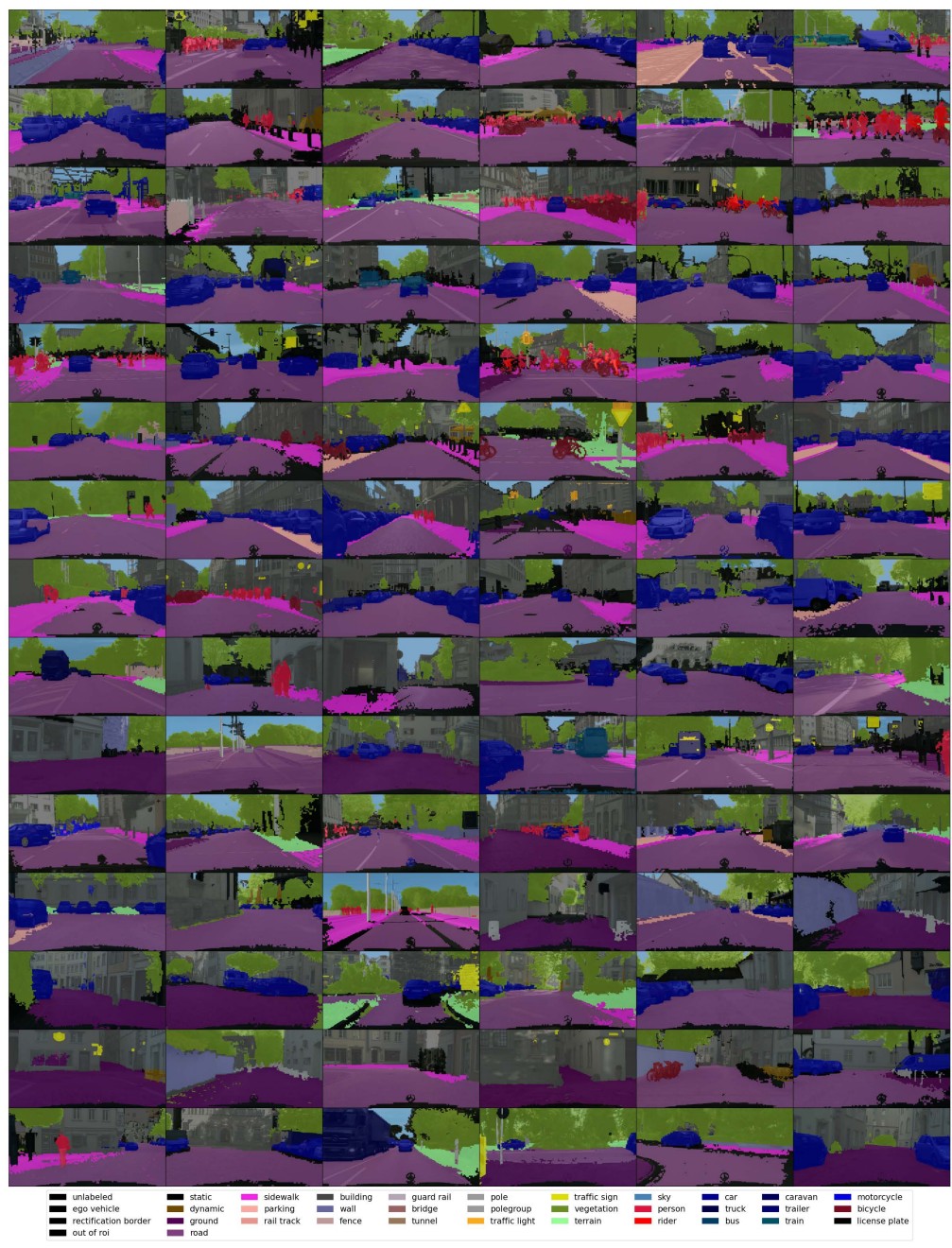

Figure 14: **Qualitative results on Cityscapes.** Random sampling from a subset of our results, refined with CRF, with NMCovering greater than 40%. We assign unsupervised masks to the best overlapping classes.

