# OpenReview forum: "Hierarchy-Agnostic Unsupervised Segmentation: Parsing Semantic Image Structure"
_NeurIPS.cc/2024/Conference — NeurIPS 2024 poster_

### Official Review · Reviewer_Xo6d · 2024-07-08

**Soundness:** 2
**Presentation:** 2
**Contribution:** 2
**Rating:** 5
**Confidence:** 3

**Summary:**

The paper presents a novel approach to unsupervised semantic image segmentation.  The authors introduce a novel algebraic methodology that constructs a hierarchy-agnostic semantic region tree, which dynamically identifies scene-conditioned primitives and creates a nuanced and unbiased segmentation of image pixels.
Key contributions of the paper include:
- A deep recursive spectral clustering method that maximizes an unbiased measure of total semantic similarity across multiple levels of semantic granularity.
- The integration of this method with diverse self-supervised learning models to enhance flexibility and applicability in unsupervised segmentation tasks.
- The introduction of new metrics, namely Normalized Multigranular Covering (NMCovering) and Normalized Hierarchical Covering (NHCovering), for estimating the quality of semantic segmentation at different levels of hierarchy.

**Strengths:**

- The paper is well-written. I like the figures in this paper.
- The motivation is clear and the idea of recursive spectral clustering is novel and make-sense.
- The paper introduces a unique algebraic approach to unsupervised semantic segmentation, offering a new perspective on hierarchical image parsing without predefined structures.
- The novel metrics proposed for evaluating segmentation quality are innovative and practical.

**Weaknesses:**

- The proposed approach has a few critical hyperparameters (the minimum number of points in a partition $k_{min}$, a threshold $p_{max}$,   a maximum eigenvalue $\lambda_{max}$) which if not properly set can degrade the performance.

**Questions:**

no

**Limitations:**

see weakness

---

> ### Author Rebuttal · Authors · 2024-08-06
>
> Dear Reviewer **Xo6d**,
>
> Thanks for your comments and your critical remarks on the parameters.
>
> We make some observations concerning them.
>
> You note that the parameters require *being well set*. However:
>
> 1) Table 6 (c) and Table 6 (a) on page 9 of the main paper show that $k_{min}$, $\\lambda_{max}$ and $p_{max}$ influence is limited, not affecting the performance significantly.
>
> 2) Parameters are helpful for *adapting* our general hierarchy-agnostic segmentation method to many datasets' semantic hierarchies, which are remarkably different.
>
> Parameters are used within the recursive partitioning to cope with the rather distinct detail levels (or granularity) in each dataset, as commented on page 18, Appendix D1, about the different properties of each dataset in terms of their parts hierarchy (e.g. *scene centric*, *object-centric*).
>
> - $𝑘_{𝑚𝑖𝑛}$ sets the minimum number of points needed to define a partition. For example, in Pascal Parts, parts can be pretty tiny, like eyebrows, which is not the case in COCO-Stuff.
>
> - $𝑝_{𝑚𝑎𝑥}$ instantiates Theorem 1 upper-bound  (see Appendix A, page 15), according to the multiplicity at each level of a dataset latent semantic hierarchy — for example, the number of car details in PascalParts as opposed to CityScape, likewise the highly varying number of categories.
>
> - $\\lambda_{max}$  evaluates the conditions to define a primitive by looking for large eigenvalues beyond Theorem 1 upper-bound, namely for a non-changing state. It adapts to the specific elementary semantic of the dataset (e.g. COCO-stuff primitives as opposed to Imagenet Parts primitives).
>
> These parameters, which are entirely general for datasets with essential differences in their semantic structure, can be refined by cross-validation.
>
> Thanks again for highlighting your concerns about the parameters. If you consider the explanations helpful, we shall improve the manuscript accordingly.

---

### Official Review · Reviewer_E2ZP · 2024-07-12

**Soundness:** 3
**Presentation:** 3
**Contribution:** 3
**Rating:** 7
**Confidence:** 3

**Summary:**

This paper addresses the problem of unsupervised hierarchy-agnostic segmentation by treating it as a graph partitioning problem. Specifically, each node in the graph represents a part, while the weighted edges, measured by similarity scores, reflect the connections between parts. The graph is recursively partitioned until a stopping criterion is met. To facilitate the graph partitioning, superpixel segmentation is initially performed to construct the graph, and boundary sharpening is applied post-partitioning as a technique to improve segmentation results.
The paper also introduces two new metrics: Normalized Multigranular Covering and Normalized Hierarchical Covering, which ensure that both foreground and background instances, as well as hierarchical inclusion, are considered. Experiments conducted on popular benchmarks demonstrate the strong capability of the proposed method as an unsupervised segmentation approach.

**Strengths:**

- The method proposed in this method is novel and well-motivated, with detailed derivation given in the supplementary.
- The experimental results are strong and achieve significant improvement compared to prior arts.
- The paper is overall well-written and the figures are clean and present the information well.
- Code is provided to facilitate the reproduction of the results.

**Weaknesses:**

- The pre-trained DINOv2-ViT-B14-REG backbone is a very strong backbone and also a strong requirement when compared to other unsupervised segmentation model, which harms the fair comparison.
- The newly introduced metrics would be more clear if they could come with an example to illustrate the computation process step by step.

**Questions:**

Please refer to the weaknesses.

**Limitations:**

Limitations and broader impact of the proposed method are discusses in the paper. No other discussions are must needed.

---

> ### Author Rebuttal · Authors · 2024-08-06
>
> Dear Reviewer **E2ZP**,
>
> We sincerely appreciate your evaluation and sharp comments. Your feedback is helpful in enhancing the quality and clarity of our work.
>
> As you indicate, we will address the weaknesses.
>
> - Answer to *The pre-trained DINOv2-ViT-B14-REG backbone is a very strong backbone and also a strong requirement when compared to other unsupervised segmentation models, which harms the fair comparison.*
>
> Below, we present Table R1, which details the ablation of the backbone architecture and the pre-training strategy used in our approach relative to the PascalVOC2012 dataset.
> On the other hand, Table 5, on page 8 of the main paper, presents a backbone ablation on PascalVOC2012. Please notice likewise that we documented quantitative results using DINO-ViT-B8, a weaker backbone than DINOv2-ViT-B14-REG, on Cityscapes, KITTI, COCO-Stuff, and Potsdam in Tables 8, 9, 10, and 11, on pages 19, 20, and 21 of the Appendix.
>
> $$
> \\small
> \\begin{array}{c}
> \\text{Table R1: \\textbf{Granularity-agnostic segmentation evaluation on the PascalVOC2012 \\textit{val} set. We used a maximum overlap heuristic for category matching in each image. }}
> \\\\
> \\text{\\textbf{We report the $\\textrm{IoU}$ category for each experiment with micro and macro averaged scores and the $\\textrm{NMCovering}$. We also include results for other pre-training strategies.}}
> \\\\[1em]
> \\begin{array}{|l|c|c|c|c|c|c|c|c|c|c|c|c|c|c|c|c|c|c|c|c|c|c|c|c|c|}
> \\hline
> \\text{Backbone} & \\text{bkgd} & \\text{airplane} & \\text{bicycle} & \\text{bird} & \\text{boat} & \\text{bottle} & \\text{bus} & \\text{car} & \\text{cat} & \\text{chair} & \\text{cow} & \\text{d. table} & \\text{dog} & \\text{horse} & \\text{bike} & \\text{person} & \\text{p. plant} & \\text{sheep} & \\text{couch} & \\text{train} & \\text{tv} & \\textrm{mIoU} & \\textrm{pAcc} & \\textrm{mAcc} & \\textrm{fIoU} & \\textrm{NMCovering} \\\\
> \\hline
> \\text{ViT-B8} \\;[1] & 63.9 & 58.5 & 40.1 & 60.5 & 58.0 & 59.7 & 74.1 & 68.6 & 68.8 & 49.7 & 67.5 & 52.0 & 65.6 & 68.6 & 58.5 & 60.5 & 58.1 & 66.5 & 62.4 & 64.2 & 52.4 & 60.9 & 69.8 & 75.1 & 63.6 & 60.8 \\\\
> \\text{CLIP-ViT-B16} \\; [2] & 74.4 & 73.0 & 52.2 & 82.0 & 71.2 & 66.5 & 76.5 & 84.0 & 87.4 & 66.4 & 86.3 & 59.1 & 83.2 & 80.3 & 75.0 & 76.1 & 70.0 & 85.5 & 79.2 & 70.5 & 63.3 & 74.4 & 79.5 & 84.0 & 75.1 & 74.0 \\\\
> \\text{MAE-ViT-B16} \\;[3] & 66.2 & 81.4 & 54.6 & 85.8 & 73.4 & 71.4 & 82.4 & 80.6 & 83.8 & 64.8 & 85.1 & 66.8 & 83.8 & 81.4 & 74.6 & 72.6 & 66.5 & 87.3 & 77.0 & 76.2 & 68.7 & 75.4 & 73.5 & 85.9 & 69.1 & 70.0 \\\\
> \\text{MOCOv3-ViT-B16} \\;[4] & 72.2 & 82.6 & \\textbf{57.2} & 83.0 & 74.4 & 69.9 & 78.7 & 76.1 & 81.8 & 59.0 & 85.7 & 66.7 & 80.3 & 77.2 & 72.3 & 70.6 & 60.2 & 86.4 & 77.6 & 76.4 & 61.8 & 73.8 & 78.1 & 84.9 & 73.0 & 71.1 \\\\
> \\text{DINO-ResNet-50} \\;[5] & 67.2 & 65.7 & 47.6 & 70.2 & 58.8 & 49.8 & 66.8 & 56.6 & 73.9 & 46.8 & 75.6 & 47.1 & 70.3 & 71.6 & 60.7 & 55.2 & 52.6 & 77.5 & 59.5 & 63.7 & 39.2 & 60.8 & 73.3 & 76.0 & 65.7 & 61.9 \\\\
> \\text{DINO-ViT-S8} \\;[5] & 69.7 & 83.1 & 51.7 & 85.8 & 75.2 & 70.2 & 84.0 & \\textbf{82.0} & 86.7 & 67.1 & 85.8 & 66.3 & 85.8 & 80.0 & 76.5 & 73.5 & 66.3 & 86.4 & 81.3 & 75.9 & 66.9 & 76.2 & 76.8 & 85.6 & 72.0 & 72.5 \\\\
> \\text{DINO-ViT-B8} \\;[5] & 70.6 & \\textbf{87.0} & 57.1 & \\textbf{91.0} & \\textbf{77.1} & 74.3 & 83.7 & 80.0 & 88.1 & 67.5 & 86.2 & 65.2 & 85.5 & \\textbf{81.2} & 78.6 & 75.0 & 66.2 & \\textbf{88.9} & \\textbf{83.5} & 80.0 & 67.6 & 77.8 & 77.4 & 86.0 & 73.0 & 74.0 \\\\
> \\text{DINOv2-ViT-B14-R} \\;[6] & \\textbf{76.9} & 73.4 & 51.0 & 82.1 & 72.4 & \\textbf{82.5} & \\textbf{85.6} & 81.1 & \\textbf{90.2} & \\textbf{71.2} & \\textbf{87.1} & \\textbf{68.8} & \\textbf{87.7} & 78.3 & \\textbf{79.2} & \\textbf{82.1} & \\textbf{70.8} & 84.7 & 82.9 & \\textbf{82.9} & \\textbf{68.8} & \\textbf{78.1} & \\textbf{82.6} & \\textbf{91.2} & \\textbf{78.1} & \\textbf{75.4} \\\\
> \\hline
> \\end{array}
> \\end{array}
> $$
>
> - Answer to *The newly introduced metrics would be more clear if they could come with an example to illustrate the computation process step by step*.
>
> Figures R1 and R2 in the above-attached PDF (see inside the frame *Author Rebuttal by Authors*) illustrate two examples of the computation process of the introduced metrics step by step: one for the NMCovering and the other for the NHCovering.  We hope the figures give a finer insight into their purpose.
>
> Thanks again for your thoughtful comments. If you find the table and figures satisfactory, please let us know so we can add them to the manuscript.
>
> **References**
>
> [1]  Dosovitskiy, A., et al. (2020). An image is worth 16x16 words: Transformers for image recognition at scale.
>
> [2] Radford, A., et al. (2021). Learning transferable visual models from natural language supervision.
>
> [3]  He, K., et al. (2021). Masked Autoencoders Are Scalable Vision Learners.
>
> [4]  Chen, X., et al. (2021). MoCo v3: Self-Supervised Learning for Visual Representation.
>
> [5] Caron, M., et al. (2021). Emerging Properties in Self-Supervised Vision Transformers.
>
> [6] Darcet, T., et al. (2023). Vision Transformers Need Registers.

---

### Official Review · Reviewer_47JN · 2024-07-12

**Soundness:** 3
**Presentation:** 2
**Contribution:** 3
**Rating:** 6
**Confidence:** 4

**Summary:**

The paper proposes a spectral-clustering approach to hierarchically segment an image, in an unsupervised fashion. The method starts from self-supervised features assigned to each pixel. Then, a recursive partitioning is obtained by minimizing a quadratic form for a given level of the hierarchy, and repeating the process until a stopping criterion is met. For faster computation, it is possible to start from superpixels. A Conditional Random Field can be applied on the boundaries to refine the prediction.

The core of the approach is the definition of a smoothness of function labelling on a graph, the minimization of which leads to a given level of the hierarchy.

Experiments show that the method achieves state-of-the-art results on several datasets.

**Strengths:**

- The idea is somehow simple, yet achieves remarkable results
- There are numerous experiments to assess the quality of the results
- Two new metrics are introduced, one granularity-agnostic, the other hierarchy-agnostic.

**Weaknesses:**

- The writing of the paper could have been simpler, and more to the point. I find that many sentences are convoluted, which makes it difficult to understand the description of the method. For example, I did not understand the method until I read the first paragraph of Appendix A.
- At a high-level, the difference of this paper with papers like [56] is only the spectral clustering method used in the process. There are other differences, but they are minor in theory.

**Questions:**

- I do not understand why the Normalized cuts (or any other spectral method) would not lead to the same kind of results. Specifically, I would like an ablation experiment in which another spectral method is used, as in [56] for example, but with the same superpixels, same network, and similar other details. The question is then: is the minimization of the smooth function-labelling the important idea here, or some other component of the approach? I believe the difference should not be as high as it appears in the paper.

**Limitations:**

The authors state that their approach can be slow, but no timing is reported.

---

> ### Author Rebuttal · Authors · 2024-08-06
>
> Dear Reviewer **47JN**,
>
> Thanks for the valuable comments and the interesting questions.
>
> - Answer to Q1.1
>
> As you fairly suggest, we report in Table R3, attached below,  *"an ablation experiment in which another spectral method is used, as in [56], ..,  with the same superpixels, network, and similar other details"*.
>
> $$
> \\small
> \\begin{array}{c}
> \\text{Table R3: \\textbf{Segmentation evaluation on PascalVOC2012 \\textit{val} set between recursive (Ours) and simultaneous deep spectral clustering (Melas-Kyriazi et Al. [6]) methods for $m=\\{4,8,16\\}$ (superpixels) using a maximum}}
> \\\\
> \\text{ \\textbf{overlap heuristic for category matching in each image. All the experiments run on pre-extracted features with DINO-ViT-S8 [5] without CRF post-processing. The other parameters in our method are}}
> \\\\
> \\text{ \\textbf{set as default in Sec. $4$ on page $7$ of the main paper. For non-hierarchical spectral clustering methods, i.e. [6], the $\\textrm{NMCovering}$ equals the Normalised Foreground Covering ($\\textrm{NFCovering}$) [7]. }}
> \\\\[1em]
> \\begin{array}{|l|c|c|c|c|c|c|c|c|c|c|c|c|c|c|c|c|c|c|c|c|c|c|c|c|}
> \\hline
> \\text{Method}&\\text{bkgd}&\\text{airplane}&\\text{bicycle}&\\text{bird}&\\text{boat}&\\text{bottle}&\\text{bus}&\\text{car}&\\text{cat}&\\text{chair}&\\text{cow}&\\text{d.table}&\\text{dog}&\\text{horse}&\\text{bike}&\\text{person}&\\text{p.plant}&\\text{sheep}&\\text{couch}&\\text{train}&\\text{tv}&\\textrm{mIoU}&\\textrm{pAcc}&\\textrm{mAcc}&\\textrm{fIoU}&\\textrm{NMCovering}\\\\
> \\hline
> \\hline
> [6]\\;(m=4)&39.4&47.7&23.6&35.6&36.1&26.4&46.5&31.6&45.6&25.3&45.1&43.2&41.8&37.2&45.7&35.5&21.0&42.5&45.5&47.0&20.6&36.4&45.5&60.0&39.3&40.7\\\\
> \\text{Ours}\\;(m=4)&54.8&34.4&13.0&23.5&25.6&20.7&50.6&25.0&48.8&18.8&40.3&29.0&36.5&40.1&39.1&31.8&15.2&34.9&36.8&41.1&18.3&32.3&62.5&68.4&49.4&44.8\\\\
> \\hline
> [6]\\;(m=8)&27.8&44.2&32.1&42.3&39.6&26.2&29.9&31.9&34.5&37.0&35.3&37.0&35.5&37.7&39.8&36.5&30.7&34.2&40.2&35.1&39.9&35.5&31.5&42.6&29.9&36.4\\\\
> \\text{Ours}\\;(m=8)&61.1&51.0&22.1&42.0&42.6&34.8&69.5&46.6&70.0&29.3&64.1&41.3&58.4&53.1&56.1&43.6&24.1&57.7&54.5&63.3&34.8&48.6&69.1&76.4&58.5&55.5\\\\
> \\hline
> [6]\\;(m=16)&16.0&28.7&33.0&29.3&30.9&23.6&17.9&24.0&21.4&33.5&22.2&28.1&22.0&24.3&25.1&27.5&31.0&23.5&29.1&21.2&34.2&26.0&19.0&27.8&18.5&27.7\\\\
> \\text{Ours}\\;(m=16)&65.2&66.8&33.7&58.7&51.9&49.8&76.1&58.8&78.1&39.4&70.0&50.8&72.2&67.3&65.2&56.9&32.9&64.3&60.2&71.6&42.5&58.6&72.1&78.9&64.3&62.2\\\\
> \\hline
> \\hline
> \\text{Ours}\\;(m=100)&69.7&83.1&51.7&85.8&75.2&70.2&84.0&82.0&86.7&67.1&85.8&66.3&85.8&80.0&76.5&73.5&66.3&86.4&81.3&75.9&66.9&76.2&76.8&85.6&72.0&72.5\\\\
> \\hline
> \\end{array}
> \\end{array}
> $$
>
> Table R3, comparing our approach to [56], shows that the fixed number of categories prior and fixed granularity prior are incompatible with the hierarchical organisation of semantic concepts and do not progress with finer superpixels (over clustering).
>
> - Answer to Q1.2
>
> As you note, minimising *smooth function-labelling* is a central idea; thanks for highlighting that.
>
> However, we remark that it operates via the eigengap, which is used to estimate the number of clusters at each tier of the recursion in an utterly general setting, namely for quite diverse dataset semantic hierarchies.
> As a consequence:
>
> 1) The number of clusters is *estimated*, exploiting perturbation and the eigengap upper bound (see Theorem 1, in Appendix A); that is, the *number of clusters* is *not defined a priori*, as opposed to other approaches resorting to spectral methods too.
>
> 2) Recursive partitioning estimates the number of clusters at each tier, accounting for a granularity *peculiar* to the variable number of parts in each dataset.
>
> 3) Finally, the novel metrics account for the hierarchical structure of the many components discovered.
>
> Perhaps these contributions to flexibility and compliance to the diverse hierarchies make a difference.
>
> Table R4 reports running time ($sec/iter$) comparisons among superpixel strategies (see Tables 6a and 6c, page 9 of the main document).
>
> \\[
> \\small
> \\begin{array}{c}
> \\text{Table R4: \\textbf{Timing w.r.t NMCCovering (left table) and NHCcovering (right table)}} \\\\[1em]
> \\begin{array}{|l|c|c|c|}
> \\hline
> \\textrm{NMCCovering}&{k_{\\min}}&k_{\\min}&k_{min}=1\\\\
> \\textbf{Superpixel}\\;(m=100)&1&5&(\\text{sec/iter})\\\\
> \\hline
> \\textit{colour-space}&&&\\\\
> \\text{k-mean}\,[1]&32.7&33.4&0.73\\pm0.14\\\\
> \\text{SLIC}\,[2]&58.1&44.9&\\textbf{0.05}\\pm0.01\\\\
> \\text{quick-shift}\,[3]&\\textbf{60.8}&\\textbf{58.0}&0.21\\pm0.12\\\\
> \\hline
> \\textit{SSL-latent-space}&&&\\\\
> \\text{k-mean}\,[1]&62.9&58.1&\\textbf{0.34}\\pm0.19\\\\
> \\text{Spectral}\,[4]&\\textbf{63.1}&\\textbf{59.9}&0.61\\pm0.11\\\\
> \\hline
> \\hline
> \\text{None}&65.7&64.9&0.84\\pm0.29\\\\
> \\hline
> \\end{array}
> \\begin{array}{|l|c|c|c|c|}
> \\hline
> \\textrm{NHCCovering}&{\\lambda_{\\max}}&{\\lambda_{\\max}}&{\\lambda_{\\max}}&\\lambda_{\\max}=0.5\\\\
> p_{\\max}&0.5&0.8&\\text{None}&(\\text{sec/iter})\\\\
> \\hline
> 50&36.9&41.1&41.4&0.49\\pm0.09\\\\
> 80&39.5&42.9&43.1&0.65\\pm0.16\\\\
> \\text{None}&40.9&43.7&44.0&0.77\\pm0.23\\\\
> \\hline
> \\end{array}
> \\end{array}
> \\]
>
> Thanks again for your effort; if you find these additions unambiguous and accurate, we add them to the manuscript.
>
> **References**
>
> [1] Lloyd, Stuart (1982). Least squares quantization in PCM.
>
> [2] Achanta, R., et Al. (2012). SLIC superpixels compared to state-of-the-art superpixel methods.
>
> [3] Vedaldi, A., et Al. (2008). Quick shift and kernel methods for mode seeking.
>
> [4] Ng, A., et Al. (2001). On spectral clustering: Analysis and an algorithm.
>
> [5] Caron, M., et Al. (2021). Emerging Properties in Self-Supervised Vision Transformers.
>
> [6] Melas-Kyriazi, et Al. (2022). Deep Spectral Methods: A Surprisingly Strong Baseline for Unsupervised Semantic  Segmentation and Localization.
>
> [7] Ke, T., et Al. (2022). Unsupervised Hierarchical Semantic Segmentation with Multiview Cosegmentation and Clustering Transformers.

---

> > ### Comment · Reviewer_47JN · 2024-08-12
> >
> > Thanks for the detailed answer.
> >
> > As I have some concerns regarding the writing of the paper (I find the paper somehow difficult to read), I maintain my previous score of weak accept.

---

### Official Review · Reviewer_oEkj · 2024-07-13

**Soundness:** 3
**Presentation:** 3
**Contribution:** 2
**Rating:** 6
**Confidence:** 3

**Summary:**

The submission presents a way to get hierarchical segmentations from the features extracted by an unsupervised semantic segmentation model. It builds a graph representation from the features or "codes" from the network. Spectral clustering is computed on this graph, followed by recursively partitioning the clusters until certain stopping conditions are met (for which the partition won't be further subdivided). It also proposes new evaluation metrics for evaluating a hierarchical segmentation against ground-truth pixel labels.

**Strengths:**

i) Variable levels of hierarchy, adaptive to the image and object being segmented.

The recursive partitioning will keep dividing the partitions into sub-partitions until certain stopping criteria are met, as described at the end of Section 3.1. A key choice is to look at the eigenvalues of the graph laplacian, to get a measure of the how well the region can be divided. So the method is seeking to find some "natural" number of levels of hierarchy in the image and its objects, without having to assume a pre-defined set of part-object relations or depth of the tree of these relations.

ii) Consistently, across multiple experiments, ablates the CRF "boundary sharpening" post-processing.

This kind of post-processing is an important thing to ablate.

iii) In experiments on pre-processing, in Table 6, pays attention to computation cost requirements.

Also runs multiple trials to get error bars on timing.

**Weaknesses:**

iv) Features are frozen.

From what I can tell, the hierarchy found by this method doesn't, and isn't meant to, play a role in the training of the feature extractor. This therefore seems likely to have a more limited impact than some related work, such as [42, 88]. Since the feature extractor could also be used as a foundation for other segmentation or vision tasks.

v) Reliance on superpixel pre-processing. This is a practical choice, but it would likely limit the method at the finest levels of the hierarchy.

**Questions:**

1. Is there some property of the proposed method that limits it to using features/codes from the self-supervised and unsupervised models described in section 2, such as DINO, STEGO and Smooseg? Or could it also be applied to codes from any arbitrary model that extracts dense image segmentation features?
2. What's the motivation for the specifics of stopping criterion #3? As in, it seems quite intuitive to look at the eigenvalues and the "smoothness" of the labelling, but why a fixed threshold? Is there any relationship to the use of the eigengap as a measure of the number of clusters?
3. Was ablating the CRF boundary sharpening also looked at for other datasets, such as Cityscapes and Mapillary Vistas?

**Limitations:**

Limitations seem clear from the authors' own description. One limitation that might follow from (iii) is that the method couldn't discover sub-parts not learned by the original feature extractor training: for example if the pixel features/codes don't differentiate a given part from its surrounding object.

---

> ### Author Rebuttal · Authors · 2024-08-06
>
> Dear Reviewer  **oEkj**,
>
> Thank you for your interesting comments and for allowing us to review some relevant points.
>
> - Answer to Q1
>
> Table R1, attached below, reports experiments ablating features extracted with different pre-training strategies, namely self-supervised [3,4,5,6] supervised classification [1], natural language supervision [2]) and backbone architectures. It extends Table 5 on page 8 of the main paper. The ablations show no apparent limit to applying our method to other deep feature extractors. When applying feature extractors, a relevant aspect is the metric used to estimate similarity among extracted features.
>
> - Answer to Q2.1
>
> In our approach, *smoothness* is a core principle that applies to the deep semantic representation of images, each with different primitive designs (tightly connected semantic components). $\\lambda_{max}$ provides a means to discern the primitives. As perturbation decreases beyond the upper bound given in Theorem 1, a large eigenvalue (< 1), in principle, approaches a steady state, hence a primitive. Table 6(c), page 9, displays ablation experiments on $\\lambda_{max}$ (likewise $p_{max})$ at varying thresholds, showing its role.
>
> - Answer to Q2.2
>
> Exactly as you say, indeed. Theorem 1, as shown in Appendix A, establishes an upper bound on the subspace size that pushes W and W' away, according to the eigengap size between the $k$-th and $k+1$-th eigenvalues. Figure 3, in Appendix A, illustrates that the size of this subspace accounts for the connected components of the graph. Hence, as you noted, the eigengap is a guide for measuring the number of clusters.
>
> - Answer to Q3
>
> Table R2, attached below, reports ablation results on the CRF applied to some datasets not mentioned in Tables 3 and 4 on page 8 of the main document.
>
> Thanks again for your questions; if you think our answers are accurate, we can improve our manuscript with your suggested revisions.
>
> $$
> \\small
> \\begin{array}{c}
> \\text{Table R1: \\textbf{Granularity-agnostic segmentation evaluation on the PascalVOC2012 \\textit{val} set. We used a maximum overlap heuristic for category matching in each image.}}
> \\\\
> \\text{\\textbf{We report the $\\textrm{IoU}$ category for each experiment with micro and macro averaged scores and the $\\textrm{NMCovering}$. We also include results for other pre-training strategies.}}
> \\\\[1em]
> \\begin{array}{|l|c|c|c|c|c|c|c|c|c|c|c|c|c|c|c|c|c|c|c|c|c|c|c|c|c|}
> \\hline
> \\text{Backbone} & \\text{bkgd} & \\text{airplane} & \\text{bicycle} & \\text{bird} & \\text{boat} & \\text{bottle} & \\text{bus} & \\text{car} & \\text{cat} & \\text{chair} & \\text{cow} & \\text{d. table} & \\text{dog} & \\text{horse} & \\text{bike} & \\text{person} & \\text{p. plant} & \\text{sheep} & \\text{couch} & \\text{train} & \\text{tv} & \\textrm{mIoU} & \\textrm{pAcc} & \\textrm{mAcc} & \\textrm{fIoU} & \\textrm{NMCovering} \\\\
> \\hline
> \\text{ViT-B8} \\;[1] & 63.9 & 58.5 & 40.1 & 60.5 & 58.0 & 59.7 & 74.1 & 68.6 & 68.8 & 49.7 & 67.5 & 52.0 & 65.6 & 68.6 & 58.5 & 60.5 & 58.1 & 66.5 & 62.4 & 64.2 & 52.4 & 60.9 & 69.8 & 75.1 & 63.6 & 60.8 \\\\
> \\text{CLIP-ViT-B16} \\; [2] & 74.4 & 73.0 & 52.2 & 82.0 & 71.2 & 66.5 & 76.5 & 84.0 & 87.4 & 66.4 & 86.3 & 59.1 & 83.2 & 80.3 & 75.0 & 76.1 & 70.0 & 85.5 & 79.2 & 70.5 & 63.3 & 74.4 & 79.5 & 84.0 & 75.1 & 74.0 \\\\
> \\text{MAE-ViT-B16} \\;[3] & 66.2 & 81.4 & 54.6 & 85.8 & 73.4 & 71.4 & 82.4 & 80.6 & 83.8 & 64.8 & 85.1 & 66.8 & 83.8 & 81.4 & 74.6 & 72.6 & 66.5 & 87.3 & 77.0 & 76.2 & 68.7 & 75.4 & 73.5 & 85.9 & 69.1 & 70.0 \\\\
> \\text{MOCOv3-ViT-B16} \\;[4] & 72.2 & 82.6 & \\textbf{57.2} & 83.0 & 74.4 & 69.9 & 78.7 & 76.1 & 81.8 & 59.0 & 85.7 & 66.7 & 80.3 & 77.2 & 72.3 & 70.6 & 60.2 & 86.4 & 77.6 & 76.4 & 61.8 & 73.8 & 78.1 & 84.9 & 73.0 & 71.1 \\\\
> \\text{DINO-ResNet-50} \\;[5] & 67.2 & 65.7 & 47.6 & 70.2 & 58.8 & 49.8 & 66.8 & 56.6 & 73.9 & 46.8 & 75.6 & 47.1 & 70.3 & 71.6 & 60.7 & 55.2 & 52.6 & 77.5 & 59.5 & 63.7 & 39.2 & 60.8 & 73.3 & 76.0 & 65.7 & 61.9 \\\\
> \\text{DINO-ViT-S8} \\;[5] & 69.7 & 83.1 & 51.7 & 85.8 & 75.2 & 70.2 & 84.0 & \\textbf{82.0} & 86.7 & 67.1 & 85.8 & 66.3 & 85.8 & 80.0 & 76.5 & 73.5 & 66.3 & 86.4 & 81.3 & 75.9 & 66.9 & 76.2 & 76.8 & 85.6 & 72.0 & 72.5 \\\\
> \\text{DINO-ViT-B8} \\;[5] & 70.6 & \\textbf{87.0} & 57.1 & \\textbf{91.0} & \\textbf{77.1} & 74.3 & 83.7 & 80.0 & 88.1 & 67.5 & 86.2 & 65.2 & 85.5 & \\textbf{81.2} & 78.6 & 75.0 & 66.2 & \\textbf{88.9} & \\textbf{83.5} & 80.0 & 67.6 & 77.8 & 77.4 & 86.0 & 73.0 & 74.0 \\\\
> \\text{DINOv2-ViT-B14-R} \\;[6] & \\textbf{76.9} & 73.4 & 51.0 & 82.1 & 72.4 & \\textbf{82.5} & \\textbf{85.6} & 81.1 & \\textbf{90.2} & \\textbf{71.2} & \\textbf{87.1} & \\textbf{68.8} & \\textbf{87.7} & 78.3 & \\textbf{79.2} & \\textbf{82.1} & \\textbf{70.8} & 84.7 & 82.9 & \\textbf{82.9} & \\textbf{68.8} & \\textbf{78.1} & \\textbf{82.6} & \\textbf{91.2} & \\textbf{78.1} & \\textbf{75.4} \\\\
> \\hline
> \\end{array}
> \\end{array}
> $$
>
> $$
> \\small
> \\begin{array}{c}
> \\text{Table R2: \\textbf{CRF ablation of our algorithm on different datasets using a maximum overlap heuristic for category matching.}} \\\\[1em]
> \\begin{array}{|l|c|c|}
> \\hline
> \\text{Dataset} (\\textrm{mIoU}) & \text{w/o CRF} & \text{w CRF} \\\\
> \\hline
> \\text{Cityscapes} & 48.8 & 51.0 \\\\
> \\text{KITTI-STEP} & 51.2 & 53.4 \\\\
> \\text{Mapillary Vistas} & 47.6 & 48.5 \\\\
> \\text{Potsdam} & 58.9 & 63.2 \\\\
> \\hline
> \\end{array}
> \\end{array}
> $$
>
> **References**
>
> [1]  Dosovitskiy, A., et al. (2020). An image is worth 16x16 words: Transformers for image recognition at scale.
>
> [2] Radford, A., et al. (2021). Learning transferable visual models from natural language supervision.
>
> [3]  He, K., et al. (2021). Masked Autoencoders Are Scalable Vision Learners.
>
> [4]  Chen, X., et al. (2021). MoCo v3: Self-Supervised Learning for Visual Representation.
>
> [5] Caron, M., et al. (2021). Emerging Properties in Self-Supervised Vision Transformers.
>
> [6] Darcet, T., et al. (2023). Vision Transformers Need Registers.

---

### Author Rebuttal · Authors · 2024-08-06

We thank the reviewers for their time, effort and valuable comments.

We are glad the reviewer found the idea "*somehow simple, yet achieves remarkable results*" (47JN); and that they found the method "*novel and well-motivated, with detailed derivation*" and "*achieves significant improvement compared to prior arts*" (E2ZP). The reviewers also found that "*the idea of recursive spectral clustering is novel*" and that "*the novel metric proposed are innovative and practical*" (Xo6d). The reviewers have also clearly appreciated that "*the method is seeking to find some 'natural' number of levels of hierarchy in the image and its objects, without having to assume a pre-defined set of part-object relations or depth of the tree of these relations*"(oEkj).

We hope our answers, equipped with tables and figures, add further insight and answer the reviewers' sharp questions.

---

### Decision · Program_Chairs · 2024-09-25

**Decision:**

Accept (poster)

**Comment:**

This paper proposes unsupervised semantic segmentation by recursive spectral clustering.

It has received 2x weak accepts, 1x borderline accept, and 1x accept.  Authors have an extensive rebuttal with additional clarification and results.  Reviewers like the unsupervised algebraic approach, new metrics for estimating the quality of the semantic segmentation at different levels of the discovered hierarchy, and performance gains.  They also point out the limitations on frozen features, reliance on superpixels, minor technical innovation, unfair comparisons with stronger backbone features, hyper-parameter sensitivity, and convoluted writing.

The AC recommends acceptance based on the consensus.   Please take all the comments and rebuttal into account to improve the final version.

~~~~~~~~~~~~~~~~~~~~~~~~
Here are some extra comments from SAC.
(1) The authors are suggested to pay much effort in the layout and font size of tables, where some results can be put into appendix.
(2) Abstract, introduction, and related work can be trimmed. If necessary, the extensive discussion on related work can be also moved into appendix.